Qu *et al. Genome Biology*    (2021) 22:104

RESEARCH

# PD-L1 lncRNA splice isoform promotes lung adenocarcinoma progression via enhancing c-Myc activity

Shuang Qu[1,2†], Zichen Jiao[3†], Geng Lu[4†], Bing Yao[5], Ting Wang[6], Weiwei Rong[1], Jiahan Xu[7], Ting Fan[8], Xinlei Sun[1], Rong Yang[1], Jun Wang[4], Yongzhong Yao[7], Guifang Xu[8], Xin Yan[9], Tao Wang[3*], Hongwei Liang[1,2*] and Ke Zen[1*]

* Correspondence:
wangtao_pumc@live.cn; hwliang@
nju.edu.cn; kzen@nju.edu.cn
†Shuang Qu, Zichen Jiao and Geng
Lu contributed equally to this work.
[3] Department of Thoracic Surgery,
Nanjing Drum Tower Hospital,
Medical School, Nanjing University
Nanjing   China
[1]State Key Laboratory of
Pharmaceutical Biotechnology,
School of Life Science, Nanjing
University, Nanjing, China
Full list of author information is
available at the end of the article

## Abstract

**Background:** Although using a blockade of programmed death-ligand 1 (PD-L1) to enhance T cell immune responses shows great promise in tumor immunotherapy, the immune-checkpoint inhibition strategy is limited for patients with solid tumors. The mechanism and efficacy of such immune-checkpoint inhibition strategies in solid tumors remains unclear.

**Results:** Employing qRT-PCR, Sanger sequencing, and RNA BaseScope analysis, we show that human lung adenocarcinoma (LUAD) all produce a long non-coding RNA isoform of PD-L1 (PD-L1-lnc) by alternative splicing, regardless if the tumor is positive or negative for the protein PD-L1. Similar to PD-L1 mRNA, PD-L1-lnc in various lung adenocarcinoma cells is significantly upregulated by IFNγ. Both in vitro and in vivo studies demonstrate that PD-L1-lnc increases proliferation and invasion but decreases apoptosis of lung adenocarcinoma cells. Mechanistically, PD-L1-lnc promotes lung adenocarcinoma progression through directly binding to c-Myc and enhancing c-Myc transcriptional activity.

**Conclusions:** In summary, the PD-L1 gene can generate a long non-coding RNA through alternative splicing to promote lung adenocarcinoma progression by enhancing c-Myc activity. Our results argue in favor of investigating PD-L1-lnc depletion in combination with PD-L1 blockade in lung cancer therapy.

## Introduction

Cancer is one of the major threats to human health worldwide and is responsible for millions of deaths annually [1]. Cancer cells have developed sophisticated mechanisms to evade immune surveillance and elimination by immune cells. One such mechanism involves the expression of various immune checkpoint molecules that suppress anti-tumor immunity. Among these intrinsic negative checkpoints, programmed death protein 1 (PD-1) and its ligand 1 (PD-L1) are a prominent pair, the blockade of which has proven to be a successful immunotherapeutic strategy to treat cancer. PD-L1, also known as B7-H1 and CD274, is

a transmembrane protein commonly expressed on the surface of tumor cells. PD-L1 specifically binds to its receptor, PD-1, which is expressed on the surface of T cells, B cells, and myeloid cells [2]. The binding of PD-L1 to PD-1 on T cells activates downstream signaling that inhibits their proliferation, cytokine generation/release, and ability to exert cytotoxic activity. Though the existence of a brake on T cell function prevents autoimmunity, many tumor cells exploit this mechanism to protect themselves from T cell-mediated elimination [3]. Commercialized PD-L1 antibodies have shown tremendous success in treating melanoma and blood cancers like leukemia and lymphoma [4, 5]. However, the efficacy of PD-L1 inhibition is limited for many cancer patients with solid tumors [6–8]. The molecular basis underlying the low efficacy of PD-L1 blockade against solid tumors is unclear, albeit several contributing factors have been suggested. First, IFNγ fails to induce the expression of PD-L1 protein in certain cancers [9, 10]. It has been shown that PD-L1 can be expressed as different forms in tumor cells, including membrane PD-L1 (mPD-L1), cellular PD-L1 (cPD-L1), and soluble PD-L1 (sPD-L1) [11, 12]. Given that only mPD-L1 can bind to T cell surface PD-1 and suppress T cell function, a low level of mPD-L1 may contribute to the poor efficacy of anti-PD-L1 antibodies. As opposed to PD-L1-positive tumors, PD-L1-negative tumors have been characterized as "cold cancer." Indeed, cold tumors have exhibited unfavorable responses to PD-L1 blockade treatment [13]. An additional factor contributing to the limited efficacy of PD-L1 blockade is a lack of anti-tumor T cells infiltrating the tumor microenvironment (TME) due to poor initial antigen presentation [14, 15]. As PD-L1 largely executes its function via targeting T cells, a low level of T cell infiltration would be associated with limited efficacy of PD-L1 blockade.

Long non-coding RNAs (lncRNAs) have recently gained attention in the field of cancer research [16–18]. As RNA transcripts of 200 or more nucleotides that are not translated to proteins, lncRNAs can be transcribed from their own promotors, from the promotors of other coding or non-coding sequences of DNA or from the enhancer sequences. Some lncRNAs are derived through alternative splicing of transcribed RNA, a process that occurs in over 90% of human multi-exon protein-coding genes [19]. Several classes of lncRNAs have been discovered on the basis of diverse parameters such as transcript length, association with annotated protein-coding genes, and mRNA resemblance among others [20]. Recent discoveries suggest that lncRNAs are relevant to cancer progression whereby lncRNAs interact with both oncogenic and tumor suppressive pathways [16]. In line with this, the expression of lncRNAs has been widely reported to be dysregulated in various human cancers [17, 18]. Despite these findings, the biogenesis and regulatory role of lncRNAs in cancer development, however, remains incompletely understood.

In the present study, we report that, in addition to PD-L1 mRNA, the PD-L1 gene in human LUAD can generate a long non-coding RNA (PD-L1-lnc) through alternative splicing. Moreover, in a similar manner to PD-L1 mRNA, PD-L1-lnc in LUAD is markedly upregulated by IFNγ. Once generated, PD-L1-lnc promotes LUAD cell proliferation and invasion but decreases cell apoptosis via directly binding to c-Myc and activating c-Myc transcriptional activity. Taken together, this study identifies the PD-L1-lnc−c-Myc axis as a novel mechanism underlying human LUAD progression.

## Results

### Generation of PD-L1 lncRNA via alternative splicing in PD-L1 protein-positive or PD-L1 protein-negative primary human LUAD and cell lines

By immunohistochemical staining, we screened more than 275 human LUAD samples and their paired distal non-cancerous tissues samples from patients registered in Nanjing Drum Tower Hospital, Nanjing University School of Medicine (Nanjing, Jiangsu, China) from 2017 to 2018 (Additional file 1: Table S1) for PD-L1 protein expression. As shown in Additional file 2: Fig. S1A and Additional file 1: Table S1, the majority of lung cancer samples were PD-L1-negative, with little or no PD-L1 protein expression, which is in line with a previous report on the PD-L1 protein expression in Chinese lung cancer patients [21]. Although the percentage of PD-L1-positive lung cancers at an advanced stage was higher than that of lung cancers at an early stage [22], a considerable number of LUAD samples at advanced stages expressed little or no PD-L1 protein (Additional file 2: Fig. S1A and Additional file 1: Table S1).

To explore the mechanism underlying the varied PD-L1 protein expression in LUADs across various stages, we assessed the PD-L1 mRNA level in PD-L1 protein-positive or PD-L1 protein-negative lung cancer tissues. To our surprise, analysis of the PCR end-product identified two bands at 198 bp and 92 bp (Fig. 1a, left). Sanger sequencing showed that the 198-bp band matched the sequence of PD-L1 mRNA (Fig. 1a, right upper), while the 92-bp band was a non-coding isoform (NR_052005.1) that lacked an alternate internal segment (hereafter referred to as PD-L1-lnc) (Fig. 1a, right lower). To confirm these results, another pair of probes was designed to detect the two missing segments (Fig. 1b, upper). The agarose gel showed two bands at 878 bp and 705 bp (Fig. 1b, lower left). The 878-bp band matched the sequence of PD-L1 mRNA (Fig. 1b, upper right; Additional file 2: Fig. S2A), while the 705-bp band matched the NR_052005.1 missing 106 nt in exon 4 and 67 nt between exons 5 and 6 (Fig. 1b, lower right; Additional file 2: Fig. S2B). Furthermore, RNA BaseScope analysis [23] was performed to examine the co-expression of PD-L1 mRNA and PD-L1-lnc in human lung cancer tissues. As shown in Fig. 1c, a clear co-existence of PD-L1 mRNA (blue dots) and PD-L1-lnc (red dots) in lung cancer tissues was observed. To accurately quantify these RNA fragments, the specific probes for PD-L1 mRNA and PD-L1-lnc were designed (Fig. 1d, upper). The PCR products of these probes (for PD-L1 mRNA and PD-L1-lnc) were confirmed by agarose gel electrophoresis (Fig. 1d, lower left) and Sanger sequencing (Additional file 2: Fig. S3A-B), respectively. The qRT-PCR assay with specific probes detected similar amounts of PD-L1 mRNA (Fig. 1d, lower middle) and PD-L1-lnc (Fig. 1d, lower right) in PD-L1 protein-positive and PD-L1 protein-negative tumor samples.

Expression of PD-L1-lnc and PD-L1 mRNA were next detected in various human lung cancer cell lines, including A549, PC9, H1975, H1650, and H1299. Western blotting and flow cytometry analysis confirmed various levels of PD-L1 expression among the LUAD cell lines (Fig. 2a-b). Similar to that in primary human LUAD tissues, both PD-L1-lnc and PD-L1 mRNA were detected in various human lung cancer cells by agarose gel electrophoresis using PD-L1 mRNA/PD-L1-lnc primer 2 (Fig. 2c) or by RNA BaseScope (Fig. 2d) and RT-PCR using PD-L1 mRNA-specific primer 3 or PD-L1-lnc-specific primer 4 (Fig. 2e).

To further investigate whether PD-L1-lnc existed in other cancers in addition to LUAD, we analyzed the RNA-seq data in the TCGA database (https://www.cancer.gov/tcga.) [24].

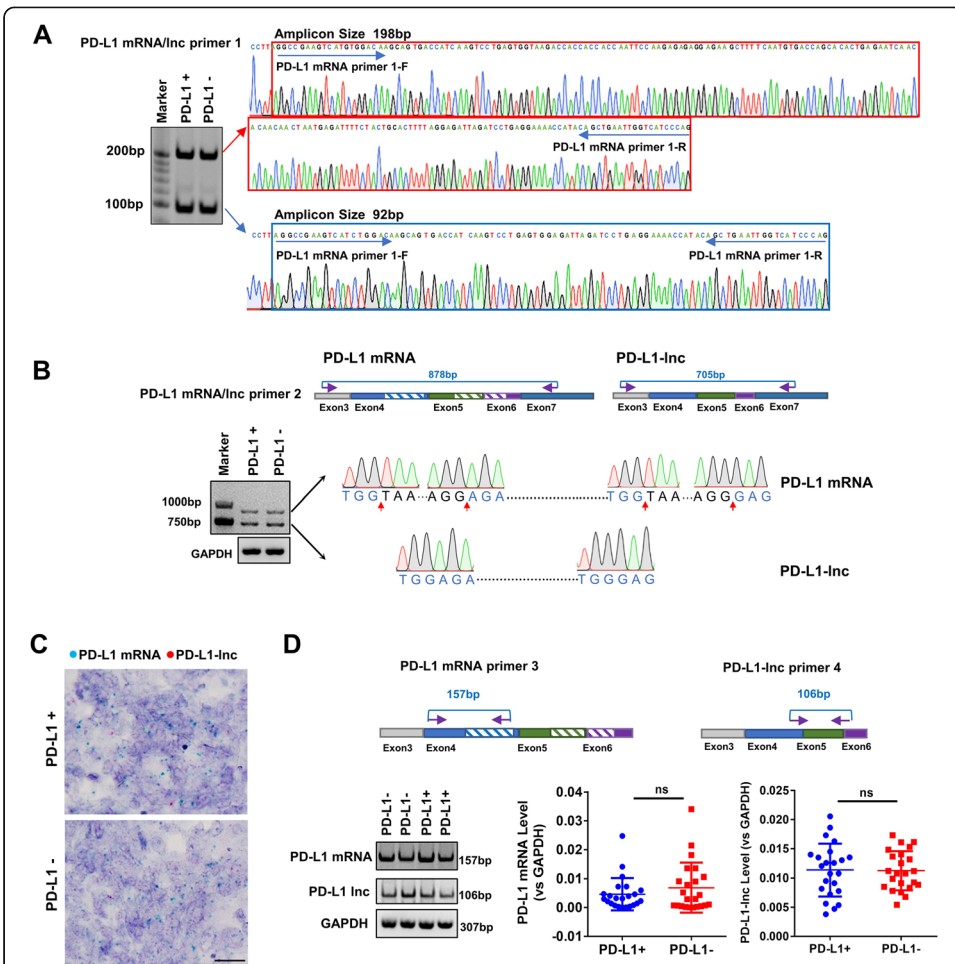

**Fig. 1** Generation of PD-L1-lnc in PD-L1 protein-positive or PD-L1 protein-negative LUAD via alternative splicing. **a** Agarose gel analysis and Sanger sequencing of the PCR end-product in PD-L1 protein-positive or PD-L1 protein-negative tumor tissues. **b** Upper, the schematic of primers for amplification of PD-L1 mRNA and PD-L1-lnc; Lower, the agarose gel and Sanger sequencing of the 878-bp and 705-bp band obtained by RT-PCR. **c** RNA BaseScope for PD-L1 mRNA (blue) and PD-L1-lnc (red) in PD-L1 protein-positive and PD-L1 protein-negative tumor tissues. Scale bar, 20 μm. **d** Upper, the schematic of primers for amplification of PD-L1 mRNA and PD-L1-lnc; Lower, the expression level of PD-L1 mRNA and PD-L1-lnc in PD-L1 protein-positive or -negative tumor tissues by qRT-PCR with specific probes. NS, no significance

The results showed that PD-L1-lnc is expressed in various cancers, including BRAC, ESCA, and STAD (Additional file 2: Fig. S4A). Pan-cancer analysis in the TCGA database further indicated a negative association between the level of PD-L1-lnc and cancer patient survival rates (Additional file 2: Fig. S4B). To confirm these results, we used specific probes to detect both PD-L1 mRNA and PD-L1-lnc in multiple cancer cell lines and primary cancer tissues (Additional file 2: Fig. S4C, left). The PCR products confirmed the expression of PD-L1-lnc in various tumor cells and tissues (Additional file 2: Fig. S4C, right).

Since PD-L1-lnc overlaps with much of PD-L1 mRNA, we determined whether PD-L1-lnc encodes for a peptide. First, we linked PD-L1-lnc to GFP mRNA in an expression system (Additional file 2: Fig. S5A, upper). RT-PCR and fluorescence microscopy showed a high level of GFP mRNA (Additional file 2: Fig. S5A, lower) but no GFP protein in the PD-L1-lnc expression system (Additional file 2: Fig. S5B). Next, we cloned

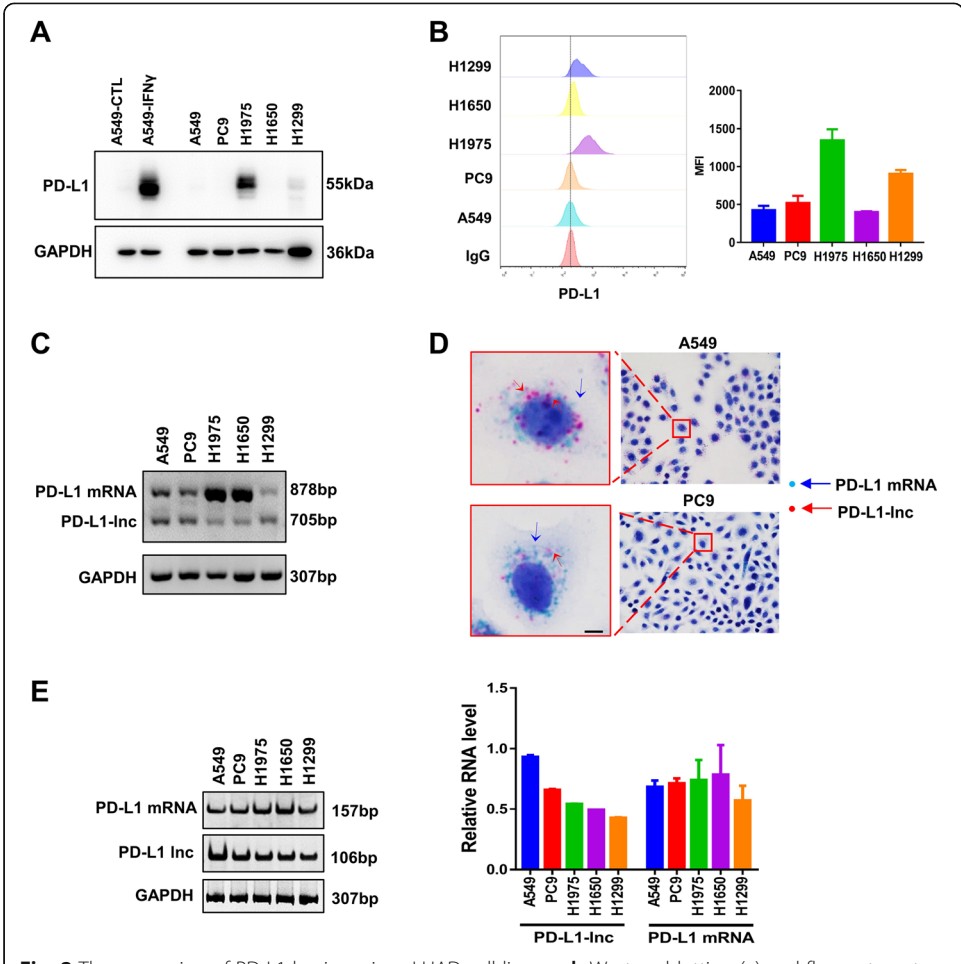

**Fig. 2** The expression of PD-L1-lnc in various LUAD cell lines. **a**, **b** Western blotting (**a**) and flow cytometry (**b**) analysis of PD-L1 in various LUAD cell lines. **c** Agarose gel analysis of PCR end-product of the PD-L1 transcript showing two bands at 878 bp and 705 bp. **d** RNA BaseScope for PD-L1 mRNA (blue) and PD-L1-lnc (red) in A549 and PC9 cells. Scale bar, 5 μm. **e** The expression level of PD-L1 mRNA and PD-L1-lnc in various LUAD cancer cell lines detected by qRT-PCR with specific primers

the PD-L1-lnc sequence into an overexpression vector (pcDNA3.1) comprising a His-tag and performed Western blotting analysis (Additional file 2: Fig. S5C, upper). As shown in Additional file 2: Fig. S5D, we did not detect the peptide using anti-His antibody. Analysis of coding status of PD-L1-lnc by NCBI ORF finder software predicts a 178aa peptide that shares the sequence of PD-L1 protein. Based on the similarity in their amino acid sequences, this peptide should be detected by anti-PD-L1 antibodies targeting the same protein region. However, no such peptide was detected by any of the several tested anti-PD-L1 antibodies (Additional file 2: Fig. S5D, upper). Taken together, these results indicate that PD-L1-lnc does not encode a peptide.

## Lung cancer cell PD-L1-lnc is markedly upregulated by IFNγ in a similar manner to PD-L1 mRNA

The expression of both PD-L1 mRNA and protein in tumor cells can be upregulated by T cell-secreted IFNγ [25], which provides a mechanism for cancer to suppress T cell

immune responses via ligating T cell surface PD-1. Given that PD-L1-lnc is spliced from 638 to 744 and 832 to 899 nucleotides downstream of the splicing site during PD-L1 mRNA transcription (Fig. 3a), we anticipated that IFNγ would enhance PD-L1-lnc transcription in a manner similar to that of PD-L1 mRNA. To test this, we treated A549, PC9, H1975, H1650, and H1299 cells with or without IFNγ followed by detection of PD-L1 protein, PD-L1 mRNA, and PD-L1-lnc levels. Western blot analysis showed that IFNγ treatment highly upregulated the expression of PD-L1 protein (Additional file 2: Fig. S6A). Increased levels of PD-L1 protein on the cancer cell surface was also detected by flow cytometry (Additional file 2: Fig. S6B). In agreement with a previous report [26], IFNγ treatment greatly increased PD-L1 mRNA levels in all tested cell lines (Fig. 3b). Similarly, IFNγ treatment significantly increased PD-L1-lnc levels across all the lung cancer cell lines (Fig. 3b). To further examine the distribution of PD-L1-lnc in lung cancer cells, we purified nuclear and cytoplasm fractions from A549 cells. As shown in Fig. 3c (left), isolated nuclear and cytoplasm fractions respectively exhibited high levels of marker proteins, Histone H3 and GAPDH, confirming the enrichment of nuclear and cytoplasm fractions in isolated products. Agarose gel (Fig. 3c, left) and RT-PCR analysis (Fig. 3c, right) demonstrated that IFNγ treatment upregulated PD-L1-lnc and PD-L1 mRNA both in nuclear and cytoplasm fractions.

## PD-L1-lnc promotes lung cancer cell proliferation and invasion but suppresses their apoptosis

LncRNAs derived from gene alternative splicing can protect the corresponding gene mRNA against the nonsense-mediated decay (NMD) pathway, which targets mRNAs

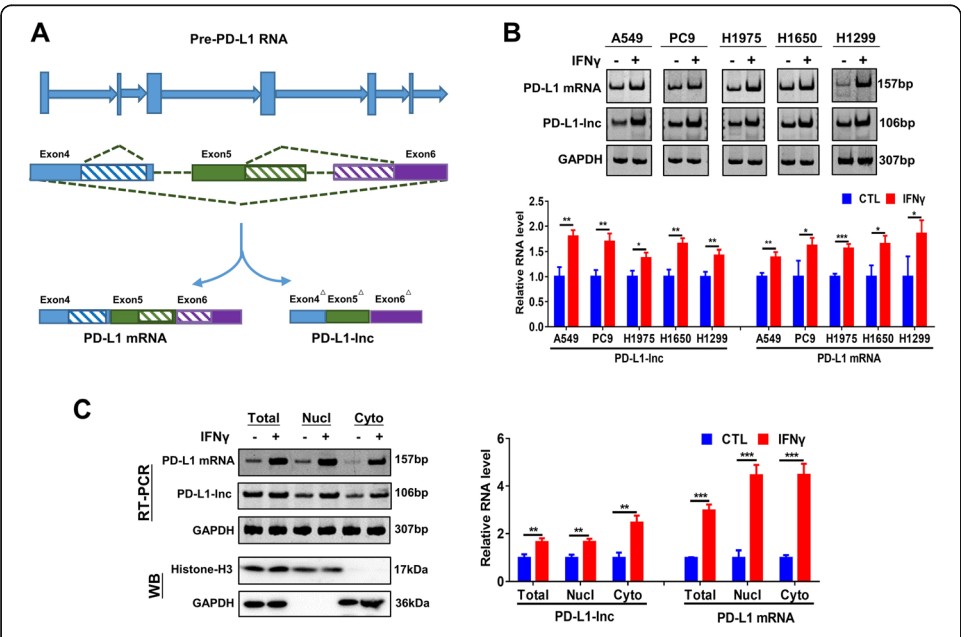

**Fig. 3** Human LUAD PD-L1-lnc is regulated by IFNγ treatment. **a** Schematic representation of the alternative splice region of the PD-L1 variants. **b** Agarose gel analysis of the expression level of PD-L1 mRNA and lncRNA in LUAD cells with or without IFNγ treatment. Upper: representative image; Lower: quantitative analysis. **c** Cellular distribution of PD-L1-lnc and PD-L1 mRNA in A549 cells with or without IFNγ treatment. Left: representative image; right: quantitative analysis. * P < 0.05; **P < 0.01; ***P < 0.001

harboring premature termination codons for degradation [27]. Since PD-L1-lnc uses the same 5′-most supported translational start codon as the PD-L1 mRNA, the PD-L1-lnc was considered a candidate for nonsense-mediated mRNA decay to protect PD-L1 mRNA. To test this, we first compared the expression level of PD-L1-lnc and PD-L1 mRNA in primary lung cancer tissues and the LUAD data set from TCGA (https://www.cancer.gov/tcga.) [24]. The results showed there was no correlation between the expression of PD-L1-lnc and PD-L1 mRNA in either our collected lung cancer tissues or the LUAD data set from TCGA (Additional file 2: Fig. S7A-B). Moreover, we also analyzed associations between PD-L1-lnc or PD-L1 mRNA and the overall survival of patients with LUAD within the TCGA database. Notably, the Kaplan–Meier survival analysis showed that LUAD patients with high PD-L1-lnc expression had shorter overall survival (Additional file 2: Fig. S7C), while the expression level of PD-L1 mRNA had no effect on their overall survival (Additional file 2: Fig. S7D). To further explore the relationship between PD-L1-lnc and PD-L1 mRNA, we either overexpressed PD-L1-lnc using a PD-L1-lnc vector or depleted PD-L1-lnc using a PD-L1-lnc shRNA in lung cancer cells and then monitored the cellular levels of PD-L1 mRNA and PD-L1 protein. As shown in Fig. S7E, the expression level of PD-L1 mRNA had no change, despite that PD-L1-lnc had been either significantly upregulated or downregulated. Western blotting and flow cytometry analyses confirmed that the overexpression or depletion of PD-L1-lnc in A549 cells has no effect on the level of PD-L1 protein (Additional file 2: Fig. S7F-G). To further monitor the effect of PD-L1-lnc on the stability or the decay of PD-L1 mRNA, A549 cells were transfected with either control vector or PD-L1-lnc vector and then were assessed by qRT-PCR after blocking the transcription of new PD-L1 mRNAs via actinomycin D (ActD) treatment. As shown in Additional file 2: Fig. S7H, there was no difference in the relative level of PD-L1 mRNA in A549 cells transfected with either control vector or PD-L1-lnc vector. Furthermore, we also investigated the effect of PD-L1-lnc on IFNγ-induced expression of PD-L1 mRNA and protein. Both flow cytometry and western blot analyses showed that PD-L1-lnc had no effect on the PD-L1 protein level in A549 cells before or after IFNγ treatment (Additional file 2: Fig. S7I-J). In contrast to our initial presumption and the prediction by the NCBI database, our data collectively demonstrate that PD-L1-lnc does not affect the expression of PD-L1 mRNA in lung cancer cells.

Although PD-L1-lnc could not enhance the expression of PD-L1 mRNA and protein, we found that it strongly affected lung cancer progression. In these experiments, cancer cell proliferation, invasion, and apoptosis were assessed using EdU staining [28], Transwell assay [29], and Annexin V labeling [30], respectively. As shown in Fig. 4, overexpression of PD-L1-lnc in A549 and PC9 lung cancer cells strongly enhanced cell proliferation (Fig. 4a) and invasion (Fig. 4b) but suppressed tumor cell apoptosis (Fig. 4c), compared to untreated cancer cells. In contrast, when PD-L1-lnc was depleted in lung cancer cells via transfection with PD-L1-lnc-specific shRNA, the proliferation (Fig. 4a) and invasion (Fig. 4b) of cancer cells were significantly reduced, whereas the apoptosis of cancer cells (Fig. 4c) was increased, compared to untreated cancer cells.

The impact of PD-L1-lnc on lung cancer cell progression was further validated in a lung cancer xenograft mouse model. For this experiment, we developed two lung cancer cell lines: A549 cells that stably express a high level of PD-L1-lnc (PD-L1-lnc) and A549 cell depletion of PD-L1-lnc (PD-L1-lnc shRNA). These modified cancer cell lines,

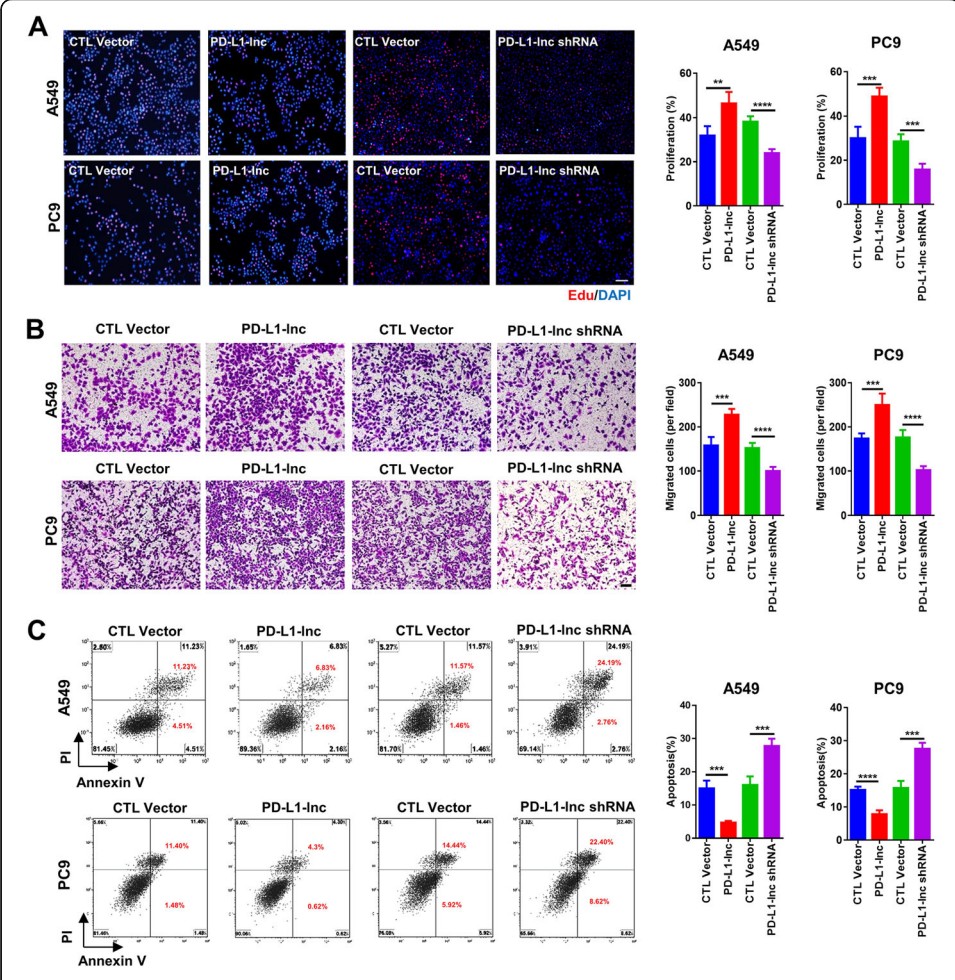

**Fig. 4** Effects of PD-L1-lnc on the proliferation, invasion, and apoptosis of human lung cancer cells. A549 and PC9 cells are transfected with PD-L1-lnc-expressing vector or PD-L1-lnc shRNA-expressing vector. **a** The proliferation of A549 and PC9 cells detected by Edu assay. **b** The invasion of A549 and PC9 cells detected by Transwell assay. **c** The apoptosis of A549 and PC9 cells detected by flow cytometry. In **a**–**c**, left: representative image; right: quantitative analysis. **$P < 0.01$; ***$P < 0.001$; **** $P < 0.0001$. Scale bars, 100 μm

along with A549 cells transfected with control vector (negative control), were subcutaneously injected into athymic nude mice. Although no significant difference in body weight was observed in the three groups of mice throughout the experiment (Fig. 5a), assessment of tumor size indicated a markedly different growth rate among these lung cancer cells (Fig. 5b–c). Specifically, A549 cells expressing a higher level of PD-L1-lnc grew significantly faster than control A549 cells, whereas A549 cells with PD-L1-lnc depleted grew slower than control A549 cells in vivo. In line with this, Ki67 staining of tumor tissue sections showed that A549 cells expressing a high level of PD-L1-lnc had a higher proliferation rate, while A549 cells with PD-L1-lnc depleted had a lower proliferation rate than control A549 cells (Fig. 5d). RT-PCR analysis of PD-L1-lnc expression in the implanted tumor tissues confirmed that PD-L1-lnc A549 tumors and PD-L1-lnc shRNA A549 tumors expressed significantly higher and lower levels of PD-L1-lnc, respectively, compared to control A549 lung cancer tissues (Fig. 5e). In contrast, the level of PD-L1 mRNA was the same between PD-L1-lnc and PD-L1-lnc shRNA A549 tumors.

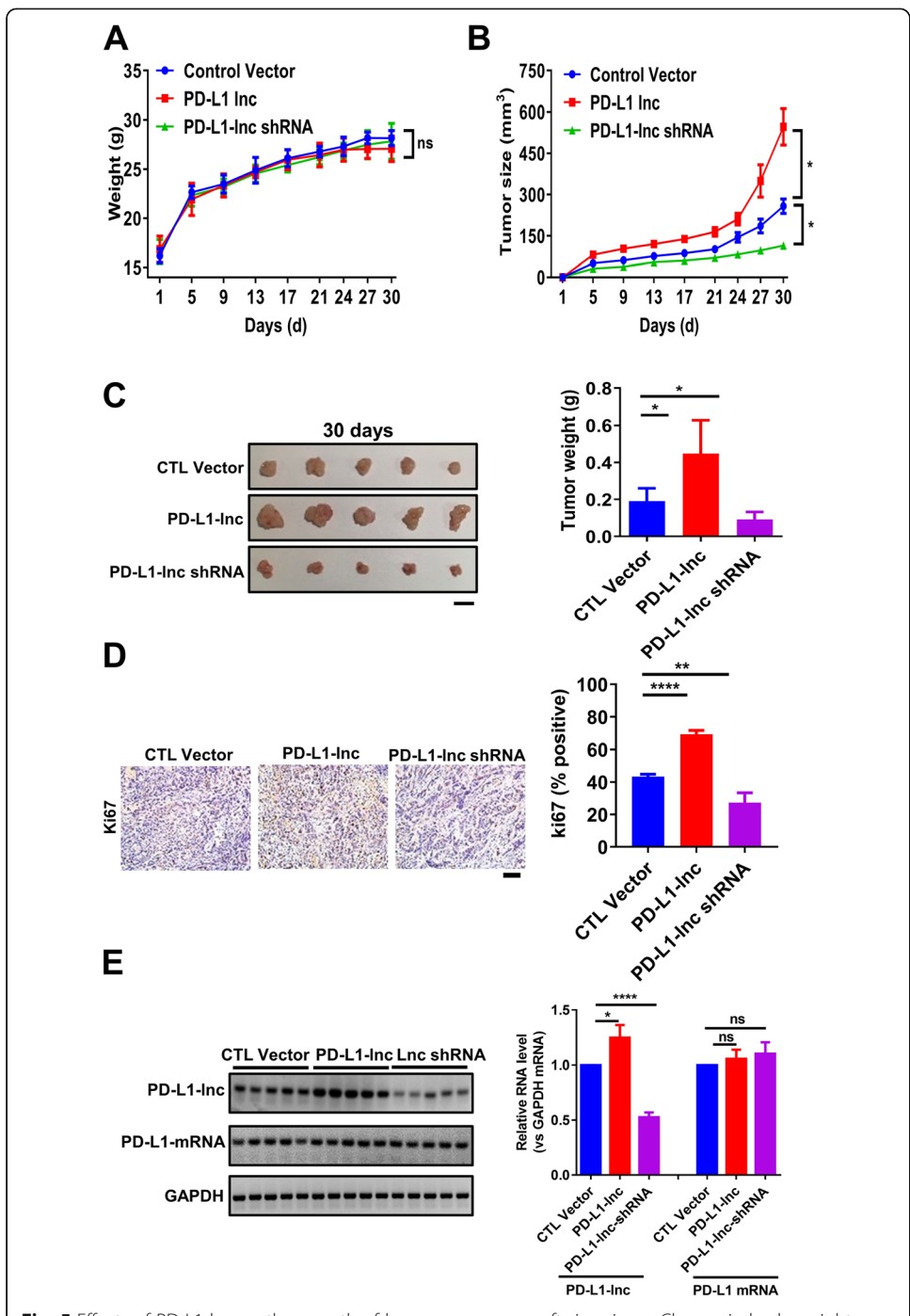

**Fig. 5** Effects of PD-L1-lnc on the growth of lung cancer xenografts in mice. **a** Change in body weight throughout the experiment. **b** Change in tumor volume. **c** The size and weight of various tumors 30 days post-engraftment. Scale bar, 1 cm. **d** Proliferation of implanted tumors. Upper: representative immunohistochemical staining of Ki67 in implanted tumors; Lower: quantitative analysis. Scale bar, 100 μm. **e** RT-PCR analysis of the expression levels of PD-L1-lnc in the implanted tumors. Upper: gel image of PCR product; Lower: RT-PCR analysis. NS, no significance; *$P < 0.05$; **$P < 0.01$; ****$P < 0.0001$

## PD-L1-lnc activates c-Myc signaling to executes its oncogenic function

To determine the mechanism underlying the cancer-promoting function of PD-L1-lnc, we performed RNA sequencing on A549 cells that were stably transfected with either

PD-L1-lnc-expressing vector (PD-L1-lnc), PD-L1-lnc shRNA, or control vector. As shown in Fig. 6a, most of the genes that were upregulated by PD-L1-lnc-expressing vector were downregulated by PD-L1-lnc shRNA vector (GEO SRA database: PRJNA684685 [31]). Further analysis of the upregulated genes by PD-L1-lnc-expressing vector or downregulated genes by PD-L1-lnc shRNA indicated that the majority of genes (~ 20%) were involved in the c-Myc signaling pathway (Fig. 6b). Next, using qRT-PCR, we examined a panel of these genes that both displayed a significant alteration as determined by RNA sequencing and were in the c-Myc-regulating network. As shown in Fig. 6c, PD-L1-lnc overexpression or depletion markedly altered these genes in both A549 cells and lung cancer xenografts (Fig. 6c and Additional file 2: Fig. S8). Moreover, we also examined the panel of these genes in A549 cells co-transfected with PD-L1-lnc overexpression vector and c-Myc siRNA vector. As shown in Fig. 6c, c-Myc siRNA significantly attenuated the promotion of the genes that were otherwise enhanced by PD-L1-lnc overexpression. These results suggested that PD-L1-lnc promotes tumor progression through c-Myc.

LncRNAs play an important role in regulating tumorigenesis by interaction with other proteins. We next employed two strategies to examine the potential interaction between PD-L1-lnc and c-Myc, a primary member of the MYC family that plays a critical role in cancer cell proliferation, metastasis, and apoptosis resistance [32]. As

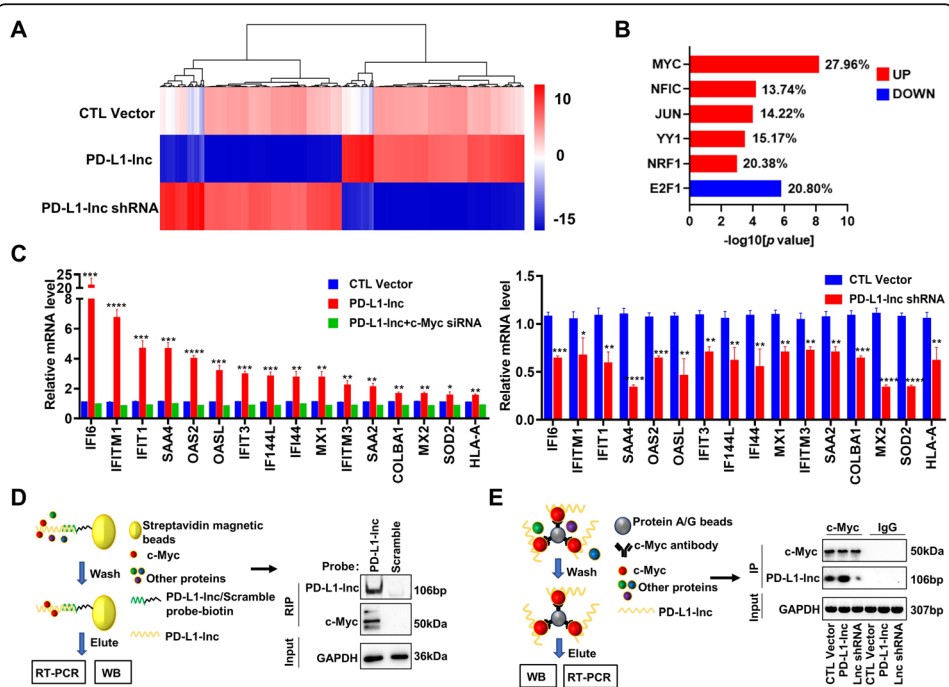

**Fig. 6** PD-L1-lnc binds to c-Myc and activates c-Myc signal downstream. **a** The heatmap of dysregulated genes in A549 cells transfected with PD-L1-lnc-expressing vector (PD-L1-lnc), PD-L1-lnc shRNA vector (PD-L1-lnc shRNA), or control vectors (CTL Vector). **b** Transcriptional factor analysis among the upregulated or downregulated genes in A549 cells following PD-L1-lnc overexpression or PD-L1-lnc knockdown. The bars reflect the percent of dysregulated genes regulated by the transcriptional factor. **c** Expression level of genes within c-Myc signaling downstream in A549 cells with PD-L1-lnc overexpression (Right) or knockdown (Left). **d** Immunoprecipitation of PD-L1-lnc in A549 cells using biotinylated antisense probes against PD-L1-lnc to detect the PD-L1-lnc-associated c-Myc. **e** Immunoprecipitation of c-Myc in A549 cells using anti-c-Myc antibody and ProteinA/G beads to detect the c-Myc-associated PD-L1-lnc. $*P < 0.05$; $**P < 0.01$; $***P < 0.005$; $****P < 0.0001$

shown in Fig. 6d (left), we designed a system consisting of biotin-conjugated antisense oligonucleotide of PD-L1-lnc and streptavidin-conjugated magnetic beads to specifically pull down PD-L1-lnc. Biotin-conjugated scramble oligonucleotide was used as a negative control. Western blot analysis showed that c-Myc was immunoprecipitated in association with PD-L1-lnc, whereas no c-Myc was detected in the immunoprecipitated complex by scramble oligonucleotide (Fig. 6d, right). We also performed similar co-immunoprecipitation experiments in MCF-7 (breast cancer cell line) and Huh-7 (liver cancer cell line), two cell lines that express no or little PD-L1 protein (Additional file 2: Fig. S9A-B). As shown in Additional file 2: Fig. S9C, c-Myc was immunoprecipitated in association with PD-L1-lnc, whereas no c-Myc was detected in the immunoprecipitated complex by scramble control oligonucleotide. Next, we immunoprecipitated c-Myc in A549 cells that were overexpressed with PD-L1-lnc or with PD-L1-lnc knockdown by shRNA, or control A549 cells, using anti-c-Myc antibody followed by Protein A/G-conjugated Sepharose beads (Fig. 6e, left). As shown in Fig. 6e (right), PD-L1-lnc was detected in the immunoprecipitated product when anti-c-Myc antibody was used, whereas no PD-L1-lnc was detected when IgG control was employed. Moreover, there was much more PD-L1-lnc in the immunoprecipitated product from PD-L1-lnc overexpressed A549 cells, and less PD-L1-lnc in the immunoprecipitated product from PD-L1-lnc knockdown A549 cells, compared to the control A549 cells. Using anti-c-Myc antibodies, we also detected PD-L1-lnc in the immunoprecipitated product from MCF-7 or Huh-7 cells (Additional file 2: Fig. S9D). Taken together, our results suggested that PD-L1-lnc associates with c-Myc in A549 cells.

The interaction between PD-L1-lnc and c-Myc was next analyzed using catRAPID omics [33] and RPISeq [34] algorithms. The interaction probabilities of PD-L1-lnc and c-Myc predicted by both algorithms were more than 0.90, and the most probable binding area of c-Myc in PD-L1-lnc was located in 500–1000 nt of PD-L1-lnc. To explore the PD-L1-lnc tertiary conformation, we first determined its secondary structures using minimum free energy algorithm implemented in Mfold (version 2.3) [35]. The results showed that PD-L1-lnc might form a hairpin structure, likely from 500 to 1000 nt. This secondary structure with a lower theoretical value of free energy was then selected as a model for 3D structure prediction. The output file containing the primary sequence and an associated secondary structure (Dot-Bracket Notation) was submitted to RNA Composer to generate the 3D structure [36]. NPDock [37] was then used to construct the in-silico molecular docking between PD-L1-lnc and c-Myc. The c-Myc 3D structure used in the docking procedure was derived from Protein Data Bank. As shown in Fig. 7a, c-Myc could bind to the double helix structure of the hairpin structure formed by PD-L1-lnc. Further analysis showed that hydrogen bonds might form between the $ASN^{100}$ of c-Myc and the $AG^{901-902}$ of PD-L1-lnc, the $GLY^{103}$ of c-Myc and the $U^{883}$ of PD-L1-lnc, the $PHE^{107}$ of c-Myc and the $G^{882}$ of PD-L1-lnc, the $ALA^{110}$ of c-Myc and the $U^{881}$ of PD-L1-lnc, the $THR^{117}$ of c-Myc and the $AU^{880-881}$ of PD-L1-lnc, the $GLU^{118}$ of c-Myc and the $G^{902}$ of PD-L1-lnc, the $GLY^{121}$ of c-Myc and the $G^{877}$ of PD-L1-lnc (Additional file 2: Fig. S10). These hydrogen bonds could enable the formation of a basic helix-loop-helix Zip motif [38] in c-Myc. To validate these predicted binding sites, two PD-L1-lnc vectors, in which the predicted binding sites were mutated (PD-L1-lnc mut) or depleted (PD-L1-lnc del), were constructed (Fig. 7b). These mutant PD-L1-lnc vectors, as well as the WT PD-L1-lnc vector, were then transfected into A549

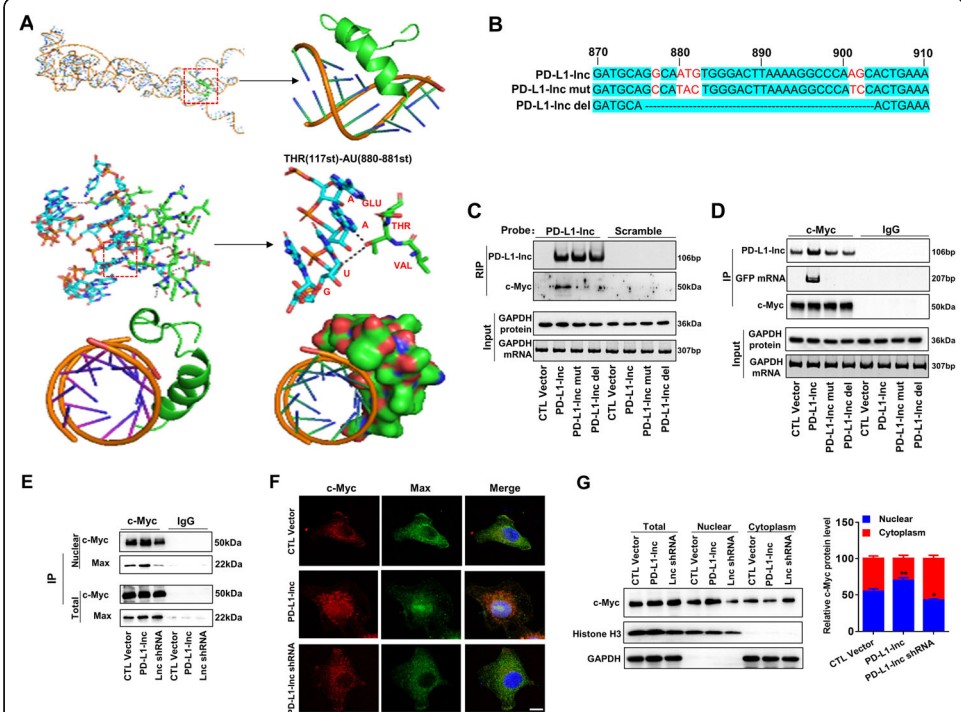

**Fig. 7** Binding of PD-L1-lnc with c-Myc enhances the association of c-Myc with Max and promotes c-Myc nuclear translocation. **a** Graphical representation of three-dimensional structures of PD-L1-lnc and c-Myc docking models with a zoom-in image of the binding interface. The binding region was shown in three different visualizations (cartoon and sphere). **b** Schematic representation of the PD-L1-lnc del vector with the predicted binding sites removed and the PD-L1-lnc mut vector which was mutated at the predicted binding sites (marked red). **c** Pulldown of PD-L1-lnc in A549 cells transfected with WT or mutant PD-L1-lnc/ GFP vector using biotinylated antisense probes against GFP mRNA to detect the binding of PD-L1-lnc with c-Myc. **d** Immunoprecipitation of c-Myc in A549 cells transfected with WT or mutant PD-L1-lnc/GFP vector using anti-c-Myc antibody to assess the binding of PD-L1-lnc with c-Myc. **e** Binding of PD-L1-lnc with c-Myc resulted in recruitment of Max by c-Myc in A549 cells. **f**, **g** Immunofluorescence (**f**) and Western blot analysis (**g**) of c-Myc distribution in A549 cells with or without PD-L1-lnc overexpression or knockdown. Note that PD-L1-lnc overexpression and PD-L1-lnc knockdown markedly increases and decreases the nuclear distribution of c-Myc in A549 cells, respectively. $*P < 0.05$; $**P < 0.01$. Scale bar, 10 µm

cells. As shown in Additional file 2: Fig. S11A-B, WT PD-L1-lnc, PD-L1-lnc del, and PD-L1-lnc mut vectors expressed PD-L1-lnc, along with its linked GFP mRNA, in A549 cells without affecting the expression of PD-L1 mRNA and protein. Immunoprecipitation assays using biotin-conjugated antisense oligonucleotide of GFP mRNA plus streptavidin-conjugated magnetic beads showed that c-Myc was associated with WT PD-L1-lnc but not PD-L1-lnc mut (Fig. 7c). In line with this, direct pulldown of c-Myc in A549 cells transfected with WT PD-L1-lnc GFP mRNA or mutant PD-L1-lnc GFP mRNA indicated that only WT PD-L1-lnc GFP mRNA but not mutant PD-L1-lnc GFP mRNA was associated with c-Myc (Fig. 7d).

Previous studies showed that the helix-loop-helix Zip motif of c-Myc enables formation of a heterodimer with the chaperone protein Max in initiating gene transcription. Given that binding of PD-L1-lnc with c-Myc can facilitate c-Myc to form a basic helix-loop-helix Zip motif (Fig. 7a), we postulated that PD-L1-lnc could enhance c-Myc transcriptional activity by enabling a c-Myc conformation change that allows for heterodimer formation with Max. To test this hypothesis, we compared the protein level of Max in the immunoprecipitated product using anti-c-Myc antibody from the lysate of

A549 cells transfected with PD-L1-lnc-expressing vector, PD-L1-lnc shRNA vector, or control vector. Cell fraction assay indicated an increase or a decrease in the nuclear distribution of PD-L1-lnc by PD-L1-lnc overexpression or PD-L1-lnc knockdown, respectively (Additional file 2: Fig. S12). As shown in Fig. 7e, compared to the A549 cells transfected with the control vector, Max protein level significantly increased in the A549 cells transfected with PD-L1-lnc-expressing vector but decreased in the A549 cells transfected with the PD-L1-lnc shRNA. Immunoprecipitation and western blot analysis further indicated that overexpression of PD-L1-lnc in A549 cells strongly enhanced the nuclear distribution of Max, which was complexed with c-Myc. In contrast, knockdown of PD-L1-lnc in A549 cells suppressed the nuclear distribution of Max. Interestingly, neither overexpression nor knockdown of PD-L1-lnc in A549 cells had an effect on the total cellular level of c-Myc-associated Max. Supporting the notion that PD-L1-lnc binds to c-Myc, immunofluorescence labeling showed that overexpression of PD-L1-lnc significantly enhanced the nuclear translocation of c-Myc in A549 cells (Fig. 7f). Cell fraction assay and western blot analysis also indicated an increase in the nuclear distribution of c-Myc by overexpression of PD-L1-lnc, whereas the total cellular level of c-Myc remained unchanged (Fig. 7g).

To further validate whether PD-L1-lnc enhances c-Myc transcriptional activity through enabling a c-Myc conformation change that allows for the predicted heterodimer formation with Max (Fig. 7a), we compared the protein level of Max in the immunoprecipitated product using anti-c-Myc antibody from the lysate of A549 cells transfected with WT PD-L1-lnc, PD-L1-lnc mut, PD-L1-lnc del, or control vectors. As shown in Additional file 2: Fig. S13A-B, immunoprecipitation and western blot analysis showed that transfection with WT PD-L1-lnc vector enhanced the nuclear distribution of Max, which was associated with c-Myc, whereas transfection with PD-L1-lnc mut or PD-L1-lnc del had no effect on the nuclear distribution of Max. Cell fraction assay and western blot analysis also indicated an increase in the nuclear distribution of c-Myc induced by WT PD-L1-lnc, whereas the nuclear distribution of c-Myc in the A549 cells transfected with PD-L1-lnc mut or PD-L1-lnc del vectors remained unchanged (Additional file 2: Fig. S13C). These results suggest that PD-L1-lnc may enhance c-Myc activity via binding to Max and promoting its entry into the nucleus.

To further test whether PD-L1-lnc executes its function through the c-Myc signaling pathway, we knocked down c-Myc expression via c-Myc siRNA in A549 cells (Additional file 2: Fig. S14), in which PD-L1-lnc was or was not overexpressed, and then monitored cell proliferation, invasion, and apoptosis. As shown in Fig. 8a–c, overexpression of PD-L1-lnc strongly promoted A549 cell proliferation (Fig. 8a), invasion (Fig. 8b), and resistance to apoptosis (Fig. 8c). Depleting cellular c-Myc via c-Myc siRNA treatment, however, largely abolished the effect of PD-L1-lnc on enhancing A549 cell proliferation (Fig. 8a), invasion (Fig. 8b), and resistance to apoptosis (Fig. 8c). The function of PD-L1-lnc mutants, which do not bind to c-Myc, was also investigated. As shown in Fig. 8a–c, transfection with PD-L1-lnc mutants did not enhance tumor cell proliferation, invasion, and resistance to apoptosis.

## Discussion

As an important immune checkpoint, PD-L1 suppresses T cell immune responses via binding to T cell surface PD-1 and initiating programmed T cell death [39, 40]. Tumor

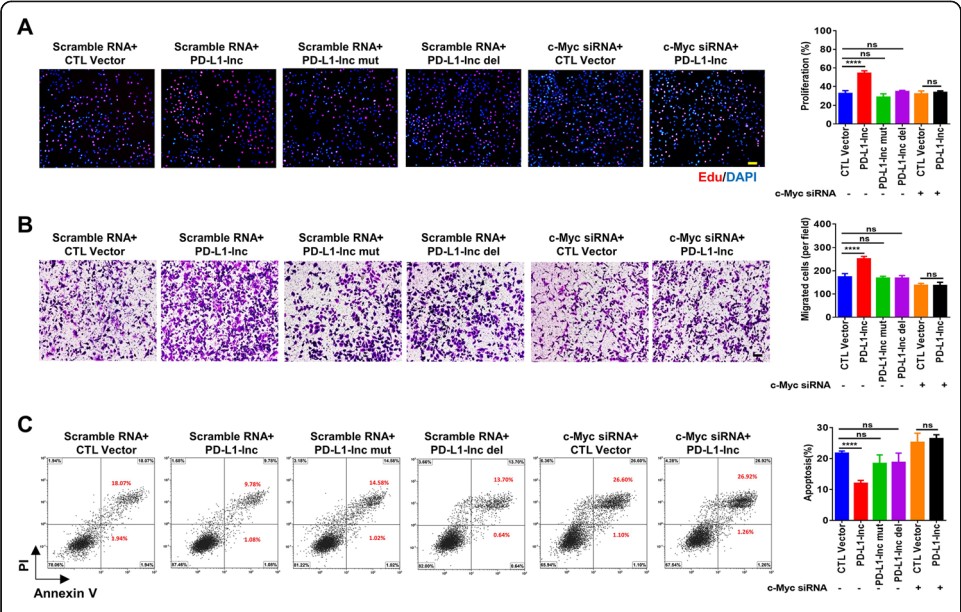

**Fig. 8** PD-L1-lnc promotes LUAD progression via c-Myc. **a–c** Overexpression of PD-L1-lnc mutant or knockdown of c-Myc protein expression in A549 cells abolished the effect of PD-L1-lnc on A549 cell proliferation (**a**), migration (**b**), and resistance to apoptosis (**c**). In A-C, left panels, representative image from three independent experiments; right panels, quantitative analysis of images. NS, no significance; ****P < 0.0001. Scale bars, 100 μm

cells often upregulate PD-L1 expression, especially in response to T cell infiltration, as a mechanism to evade T cell-mediated elimination. Blockade of PD-L1/PD-1 has thus been widely applied in tumor immunotherapy and has achieved great success in treating various cancers, particularly melanoma, leukemia, and lymphoma. However, the efficacy of PD-L1 blockade strategies is limited for solid cancers [6–8]. In the present study, we demonstrate that the PD-L1 gene in primary LUAD tissues, as well as various lung cancer cell lines, produces another PD-L1 transcript, PD-L1-lnc, via PD-L1 gene alternative splicing. In a similar manner to PD-L1 mRNA, PD-L1-lnc level is markedly increased by IFNγ. Moreover, PD-L1-lnc strongly promotes tumor cell proliferation, survival and invasion via enhancing c-Myc activity. Given that the production of PD-L1-lnc in lung cancer cells is not dependent upon PD-L1 protein expression and that depletion of PD-L1-lnc markedly suppresses tumor growth both in vitro and in vivo, depleting tumor cell PD-L1-lnc may provide a novel anti-tumor therapeutic approach in addition to PD-L1 blockade.

The efficacy of immune checkpoint blockade depends on the infiltration and activation of anti-tumor T cells, which subsequently induce significant elevation of IFNγ in the TME [41]. Although IFNγ has been regarded as an anti-tumor cytokine [42, 43], recent studies have demonstrated that IFNγ has a 'double-edge' effect in tumor progression [44, 45]. For example, IFNγ can upregulate immune checkpoint molecules such as PD-L1, which enhances tumor immune evasion [25, 46]. Here we showed that, in addition to PD-L1 mRNA, IFNγ also enhanced the expression of PD-L1-lnc. According to the structure of the PD-L1 gene, PD-L1-lnc and PD-L1 mRNA appear to be up-regulated by IFNγ through a similar mechanism.

Previous studies have reported that the expression of PD-L1 is modulated by alternative splicing [11, 47]. Zhou et al. identified four PD-L1 splicing variants that lack the

transmembrane domain, and these secretory forms of PD-L1 can also suppress the activities of both CD4 and CD8 T cells [48]. In our study, we found that the human PD-L1 gene in various lung cancer tissues and cell lines produced a non-coding isoform, NR_052005.1 or PD-L1-lnc, which was missing 106 nt in exon 4 and 67 nt between exon 5 and 6 by alternative splicing (Fig. 3a). Since this isoform lacks an alternate internal segment and uses the 5′-most supported translational start codon as used in mRNA, it may serve as a candidate for nonsense-mediated mRNA decay in protecting the degradation of mRNA (https://www.ncbi.nlm.nih.gov/gene/29126). However, we did not find that PD-L1-lnc had any effect on the levels of PD-L1 mRNA and protein; instead, it acts as a functional lncRNA that promotes LUAD progression via enhancing c-Myc activity (Fig. 9).

Compared to the coding genes, lncRNAs are usually less abundant. However, despite low expression level, lncRNAs have been shown to play an important role in various biological processes. For example, Kumar et al. reported that the lncRNA PILAR1, the average expression level of which is about 5 FPKM in LUAD, promoted cancer cell growth, migration, and suppressed chemosensitivity of etoposide [49]. Characterizing the long intergenic non-coding RNAs landscape in lung cancer using publicly available transcriptome sequencing data in TCGA, White and co-workers found that the expression level of most lncRNAs in LUAD was less than 10 FPKM [50]. They also identified lncRNA LCAL1, the average expression level of which was about 6 FPKM in the LUAD, promoted cancer cell proliferation [50]. In line with these previous findings, we found that the expression level of PD-L1-lnc in the LUAD cell line, A549, was about 10 FPKM. Of course, since PD-L1-lnc is upregulated by IFNγ, the level of PD-L1-lnc is likely greater in LUAD under the real pathophysiologic condition. However, despite its relatively low level of expression, our results demonstrated that PD-L1-lnc significantly promoted tumor growth both in vitro and in vivo as a functionally active lncRNA.

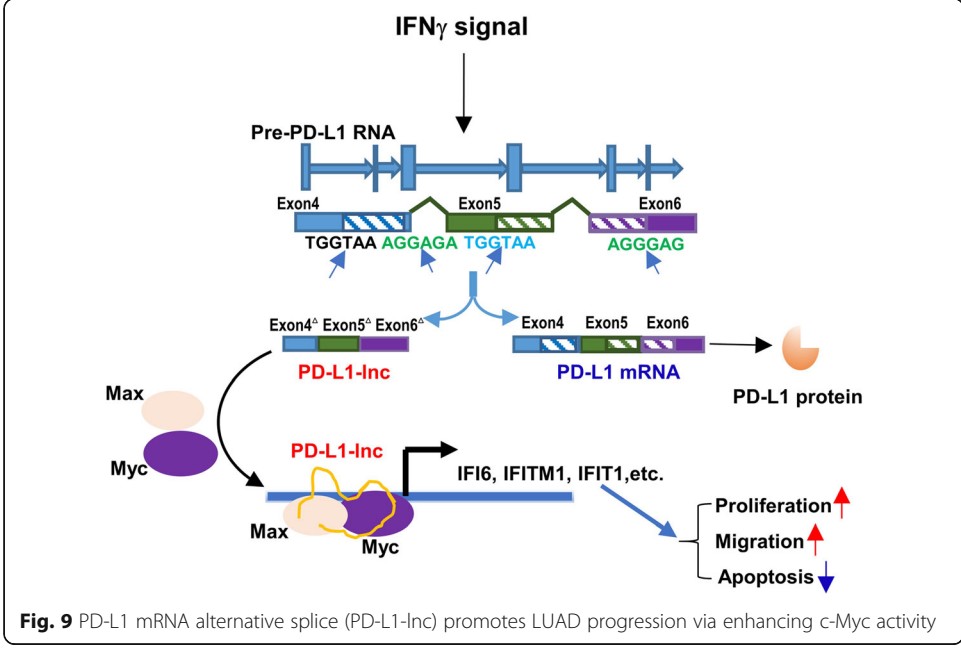

**Fig. 9** PD-L1 mRNA alternative splice (PD-L1-lnc) promotes LUAD progression via enhancing c-Myc activity

As a member of the transcription factor MYC gene family, c-Myc plays a critical role in the development of human tumors, and its overexpression has been detected in lung cancers of different histologic subtypes [51]. A previous study by Tang et al. [52] showed that lncRNA-GLCC1 could stabilize c-Myc, thereby attenuating c-Myc's ubiquitination and consequently promoting colorectal carcinogenesis. In contrast to their study in which lncRNA-GLCC1 protected c-Myc from ubiquitination and degradation through interacting with HSP90 chaperone, our data suggest that PD-L1-lnc directly interacts with c-Myc. Cross-immunoprecipitation assays clearly showed the association of PD-L1-lnc with c-Myc (Fig. 6d, e). In line with the notion that PD-L1-lnc enhances c-Myc translocation into the nucleus, and thus increases its transcriptional activity, we found that PD-L1-lnc overexpression increased the expression of a panel of c-Myc-modulated genes including IFITM1/3, MX1/2, IFI6, and SAA4. Conversely, PD-L1-lnc depletion significantly decreased the expression of these genes.

The interactions between PD-L1-lnc and c-Myc were further validated by bioinformatics analysis using catRAPID omics [33] and RPISeq algorithms [34]. The results suggested that PD-L1-lnc could form hairpin structures, leading to a partial double helix structure. Multiple amino acids of c-Myc could form hydrogen bonds with their corresponding nucleobases in the double helix structure of PD-L1-lnc, which in turn promotes c-Myc to form a basic helix-loop-helix Zip motif (Fig. 7a) and leads to the formation of more c-Myc-Max heterodimer and higher c-Myc transcriptional activity.

We have also explored the potential mechanism that modulates alternative splicing for PD-L1-lnc generation. Given that the sequences of the splice site in exon 4 and between exon 5 and 6 are UGGUAA, AGGAGA/AGGGAG, respectively, we blasted these sequences in the Tomtom database [53] and found that the sequences of these splice sites matched the motifs recognized by three alternative splicing regulators, MSI1, DAZAP1, and ESRP2 (Additional file 2: Fig. S15A). DAZAP1 was selected to further study since the expression levels of MSI1 and ESRP2 in A549 cells were too low to feasibly be functional (Additional file 2: Fig. S15B). To test whether DAZAP1 was involved in the generation of PD-L1-lnc, we checked the expression level of PD-L1-lnc in A549 cells after they were transfected with siRNAs specific to DAZAP1. As shown in Additional file 2: Fig. S15C, the expression level of PD-L1-lnc was significantly decreased (right) after the downregulation of DAZAP1 caused by siRNAs (left), arguing that DAZAP1 may control the generation of PD-L1-lnc. Further study, however, is required to clarify whether and how DAZAP1 modulates PD-L1-lnc biogenesis.

## Conclusions

The present study reveals that PD-L1-lnc, an alternatively spliced product of the PD-L1 gene, promotes LUAD proliferation, metastasis, and survival via enhancing c-Myc transcriptional activity, and also argues in favor of investigating PD-L1-lnc depletion in combination with PD-L1 blockade in lung cancer therapy.

## Methods

### Clinical samples

The 275 pairs of LUAD tissues and adjacent non-cancerous tissues were collected from patients who were diagnosed with LUAD at the Nanjing Drum Tower Hospital

(Nanjing, China) and who had not yet received treatment. The clinic pathological features are described in Additional file 1: Table S1. Tissues were collected after surgical resection and stored in liquid nitrogen before use. The study was authorized by the Ethics Committee of the Nanjing Drum Tower Hospital. All experiments were performed in accordance with relevant guidelines and regulations.

### Cell lines, culture conditions, and IFNγ treatment

Human lung cancer cell lines A549, PC9, H1975, H1299, H1650, Huh-7, and MCF-7 were purchased from American Type Culture Collection (ATCC) (Manassas, VA, USA). Cells were cultured in Dulbecco's modified Eagle's medium (DMEM, Gibco, Carlsbad, CA, USA) supplemented with 10% (v/v) FBS (Gibco), 1% (v/v) penicillin/streptomycin (Gibco). For IFNγ treatment, cells were seeded into six-well plates on day 1, targeting 70–80% of confluence on the day of surface staining. On day 2, cells were exposed to 100 ng/mL IFNγ (PeproTech, USA) for 24 h.

### Immunohistochemical staining

Tissues were fixed in 4% paraformaldehyde, embedded in paraffin, and cut into 4-μm sections. Immunohistochemical staining of PD-L1 was performed using the anti-PD-L1 antibody (SP142, Zhongshanjinqiao, China) according to the manufacturer's protocol.

### Cellular fractionation assay

The separation of nuclear and cytoplasmic fractions from cells was performed using PARIS kits (Cat#: AM1921; Life Technologies, Carlsbad, CA, USA) according to the manufacturer's instructions. Briefly, A549 cells were collected in a centrifuge tube and then washed twice using ice-cold PBS. The cells were divided in two parts, one part was lysed with ice-cold Cell Disruption Buffer to obtain total fraction. Then, the equal volume Cell Fractionation Buffer was added into the other part and the mixture was kept on ice for 5 min. Subsequently, the mixture was centrifuged at $500 \times g$ for 5 min at 4 °C. The supernatants were cytoplasmic fraction, and the pellets were lysed with ice-cold Cell Disruption Buffer to obtain nuclear fraction.

### RNA extraction and RT-PCR/qRT-PCR assay

Total RNAs were isolated by TRIzol™ Reagent (Invitrogen, USA) according to the manufacturer's instruction. The quality and quantity of RNA was assayed by a Nanodrop 2000 spectrophotometer (Thermo Fisher Scientific, USA). The nuclear and cytoplasmic fractions were purified by PARIS Kit (Ambion, Life Technologies, USA). RNA was reverse transcribed by HiScript II Q RT SuperMixfor qPCR (+gDNA wiper) (Vazyme, Nanjing, China). The 1.1×T3 Super PCR Mix (TsingKe, Nanjing, China) was used for PCR. The 10% vertical polyacrylamide electrophoresis was performed to observe the cDNA PCR products. AceQ qPCR SYBR Green Master Mix (Vazyme) was used for qRT-PCR, and GAPDH was used to normalize the level of PD-L1-lnc and PD-L1 mRNA. Primers were listed in Additional file 1: Table S2. The purified fragments were cloned into the pCR® II TA vector using the TA cloning kit (Thermo Fisher Scientific) and sequenced at the TsingKe Biotechnology to validate the cDNA PCR products.

### RNA BaseScope

BaseScope™ Probe for PD-L1-lnc and PD-L1 mRNA were designed and synthesized by Advanced Cell Diagnostics (Cat. No.700001 and 700001-C2, CA, USA). Tissues and cells were fixed by 10% neutral buffered formalin on slides for detection of PD-L1-lnc and PD-L1 mRNA. The signals of the PD-L1-lnc and PD-L1 mRNA probes were detected by BaseScope™ Detection Reagent Kit (Advanced Cell Diagnostics) according to the manufacturer's instructions. The images were acquired on Leica SP5 Scanning Laser Confocal Microscope (Leica Microsystems, Wetzlar, Germany).

### Flow cytometry

Flow cytometry was performed using a CytoFLEX S (Beckman Coulter Life Sciences, Mississauga, ON). Human LUAD cell lines A549, PC9, H1299, H1650, and H1975 were stained with 0.2 μg of PE-conjugated anti-PD-L1 (Biolegend, USA) antibody.

### Western blotting

The protein extraction reagent (Thermo Scientific) with a cocktail of proteinase inhibitors (Roche Applied Science, Switzerland) was used to isolate the total protein from cells or tissue samples. Equal amount of total protein was separated by 10% SDS–PAGE and transferred onto a PVDF membrane. Then, the membranes were blocked with 5% skimmed milk and incubated the membranes with primary antibodies at 4 °C overnight and then incubated with secondary antibodies at room temperature for 2 h. The bands were examined by Immobilob™ Western Chemiluminescent HRP Substrate (Millipore, Billerica, MA, USA). The primary antibody and secondary antibodies detailed information list below: PD-L1 [E1L3N®] XP® Rabbit mAb (#13684, Cell Signaling Technology, Beverly, MA, USA), PD-L1 mouse mAb (66248-1-Ig, Proteintech, Wuhan, China), GAPDH (6C5) Mouse mAb (sc-32233, Santa Cruz, USA), Histone-H3 Rabbit Polyclonal antibody (17168-1-AP, Proteintech), c-Myc Rabbit Polyclonal antibody (10828-1-AP, Proteintech), c-Myc (9E10) mouse mAb (Santa Cruz, sc-40), MAX antibody [EPR19352] (ab199489, Abcam, USA), His-tag mouse mAb (66005-1-Ig, Proteintech), Rabbit (DA1E) mAb IgG XP® Isotype Control (#3900, Cell Signaling Technology) and the secondary antibodies (goat anti-rabbit IgG-HRP, sc-2030, Santa Cruz; goat anti-mouse IgG-HRP, sc-2005, Santa Cruz; Mouse Anti-rabbit IgG-HRP [L27A9] mAb, #5127, Cell Signaling Technology).

### Cell proliferation assay

To measure the proliferation rate of human lung cancer cells, EdU assays were performed. Briefly, A549 and PC9 cells were seeded in 6-well plates and transfected with vectors. At 24 h after transfection, cells were harvested and reseeded 48-well plates for EdU assays. The EdU assay kit (RiBoBio, China) was used to determine the proliferation rate of cells according to the manufacturer's instructions.

### Cell invasion assay

Firstly, the upper chamber of Transwell (Millipore) was coated with diluted Matrigel (200 mg/mL, BD Biosciences, MA) at the density of 50 μL/well before use. A549 and PC9 cells were transfected with vectors. After 24 h, cells were resuspended in FBS-free DMEM medium and reseeded on the upper surface of 24-well plates. Cells were

allowed to migrate across the 8-μm membrane toward medium with 20% FBS for 24 h. Then, the cells were fixed with 4% paraformaldehyde and dyed with 0.5% crystal violet. Non-migrating cells were removed using a cotton swab. The migrant cells were blindly counted under a light microscope (Leica Microsystems, Wetzlar, Germany).

### Cell apoptosis assay

The apoptosis of A549 and PC9 cells was assayed using the Annexin V-Alexa Fluor 647/PI (YEASEN, China) based on the procedures provided by the manufacturer. The transfected A549 and PC9 cells were cultured in serum-free DMEM for 24 h. The collected cells were washed with cold PBS and resuspended in 1× binding buffer, followed by staining with Alexa Fluor 647-Annexin V and propidium iodide (PI) in the dark for 15 min. The apoptotic cells were calculated using flow cytometer (Beckman Coulter Life Sciences).

### Xenograft assays in nude mice

All animal care and handling procedures were performed in accordance with the National Institutes of Health's Guide for the Care and Use of Laboratory Animals. Male athymic BALB/c nude mice (6 weeks old) were purchased from the Model Animal Research Center of Nanjing University (Nanjing, China) and were randomly divided into 3 groups (5 mice per group). A549 cells were transfected with control vector, vector expressing PD-L1-lnc (PD-L1-lnc) or PD-L1-lnc shRNA (PD-L1-lnc shRNA) by Lipofectamine 3000 (Invitrogen) according to the manufacturer's instruction, respectively. For A549 cells transfected with control vector and PD-L1-lnc (PD-L1-lnc)-expressing vector, cells were treated with 500 μg/mL G418 (Thermo Fisher Scientific) for 2 weeks. For A549 cells transfected with PD-L1-lnc shRNA (PD-L1-lnc shRNA)-expressing vector, cells were treated with 5 μg/mL Puromycin (Thermo Fisher Scientific) for 2 weeks. Three stable lung cancer cell lines were then subcutaneously injected into mice ($10^6$ cells/0.1 ml PBS per mouse). The needle was inserted into the armpit of the left foreleg at a 45° angle and a 5-mm depth, midway down. Then, the longest diameter (a) and the shortest diameter (b) of the tumor were measured using digital calipers every 3 days, and the tumor volume (V) was calculated according to formula: $V = a \times b^2/2$. Mice were sacrificed and photographed 15 days post-injection. The xenograft tumors were removed and analyzed. Tissues were subjected to extraction of total RNA, as well as Ki67 immunohistochemical staining.

### Vector construction and cell transfection

To overexpress PD-L1-lnc, the full-length cDNA of PD-L1-lnc was synthesized and cloned into pcDNA3.1-P2A-eGFP vector (GenScript, China). To suppress PD-L1-lnc, the PD-L1-lnc shRNA vectors were synthesized and then cloned into pLKO.1 vector (GenScript). The siRNA target sequences were listed in Table S3. Cells were transfected using Lipofectamine 3000 (Invitrogen) according to the manufacturer's instruction.

### RNA sequencing and primary analysis

Total RNAs were purified from A549 cells transfected with PD-L1-lnc, PD-L1-lnc shRNA, or control vectors using TRIzol reagent and RNeasy Mini Kit (Qiagen). RNA

samples were sequenced by Illumina Novaseq™ 6000 (LC-Bio Technology Co) according to the manufacturer's instructions. Briefly, the purity and level of each isolated RNA sample was quantified using NanoDrop ND-1000 (NanoDrop, Wilmington, DE, USA). The RNA integrity was assessed by Bioanalyzer 2100 (Agilent, CA, USA) with RIN number > 7.0, and confirmed by electrophoresis with denaturing agarose gel. Poly (A) RNA was purified from 1 μg total RNA using Dynabeads Oligo (dT) 25-61005 (Thermo Fisher Scientific) with two rounds of purification. Then the poly(A) RNA was fragmented into small pieces using Magnesium RNA Fragmentation Module (NEB, cat. e6150) under 94 °C for 5–7 min. The cleaved RNA fragments were then reverse-transcribed to create the cDNA by SuperScript™ II Reverse Transcriptase (Invitrogen, cat. 1,896,649), which were used to synthesize U-labeled second-stranded DNAs with *E. coli* DNA polymerase I (NEB, cat. m0209), RNase H (NEB, cat. m0297), and dUTP Solution (Thermo Fisher Scientific, cat. R0133). Next, an A-base was then added to the blunt ends of each strand, preparing them for ligation to the indexed adapters. Each adapter contained a T-base overhang for ligating the adapter to the A-tailed fragmented DNA. Single- or dual-index adapters were ligated to the fragments, and size selection was performed with AMPureXP beads. After treating the U-labeled second-stranded DNAs with UDG enzyme (NEB, cat. m0280), the ligated products were amplified by PCR under the following conditions: initial denaturation at 95 °C for 3 min; 8 cycles of denaturation at 98 °C for 15 s, annealing at 60 °C for 15 s, and extension at 72 °C for 30 s followed by 72 °C for 5 min. The average inserting size for the final cDNA library was 300 ± 50 bp. Finally, we performed the 2 × 150 bp paired-end sequencing (PE150) on an Illumina Novaseq™ 6000 (LC-Bio Technology) according to the manufacturer's instructions. To calculate the expression of the transcripts, Cutadapt software (https://cutadapt.readthedocs.io/en/stable/, version: cutadapt-1.9) was used to remove the reads that contained adaptor contamination. After removal of the low-quality and undetermined bases, HISAT2 software (https://daehwankimlab.github.io/hisat2/, version: hisat2-2.0.4) was used to map reads to the genome. The mapped reads of each sample were assembled using StringTie (http://ccb.jhu.edu/software/stringtie/, version: stringtie-1.3.4d.Linux_x86_64) with default parameters. All transcriptomes were then merged to reconstruct a comprehensive transcriptome using gffcompare software (http://ccb.jhu.edu/software/stringtie/gffcompare.shtml, version: gffcompare-0.9.8.Linux_x86_64). After the final transcriptome was generated, StringTie and ballgown (http://www.bioconductor.org/packages/ release/bioc/html/ballgown.html) were used to estimate the expression levels of all transcripts by calculating FPKM (FPKM = [total exon fragments / mapped reads (millions) × exon length (kB)]). The differentially expressed mRNAs were selected with fold change > 2 or fold change < 0.5 by R package edgeR (https://bioconductor.org/ packages/release/bioc /html/edgeR.html).

### Streptavidin pull-down of PD-L1-lnc and c-Myc protein

For each pull-down sample, 100 μl of streptavidin magnetic beads (S1420S, NEB) were washed with wash/binding buffer (0.5 M NaCl, 20 mM Tris-HCl, pH 7.5, 1 mM EDTA) twice, then incubation with probes for 1 h at 4 °C. For biotin-coupled RNA capture, the 5′-end biotinylated PD-L1-lnc probe or control RNA were used. Probes used in these experiments were listed as follows: PD-L1-lnc probe: 5′-3′-CATCCATCATTCTC

CCAAGTGAGTCCT; GFP probe: 5′-3′-TGAAGTTCACCTTGATGCCGTT CTTC TGCTTGTCGGCCATGATATAGACGTTGTGGCTGT. The 0.5 ml cell lysis buffer (Invitrogen) with complete protease inhibitor cocktail (Roche Applied Science, IN, USA) and Recombinant RNase Inhibitor (Takara, Japan) were added into the cell pellets, and lysed by sonication. The cell lysates incubation with RNA-coupled beads followed by centrifugation at 13,000 rpm for 20 min at 4 °C. After rotating for 4 h at 4 °C, the beads were washed with 1 ml lysis buffer (containing 300 mM NaCl) twice, 1 ml low-salt lysis buffer (containing 150 mM NaCl). Half of beads were resuspended in TRIzol™ Reagent for detecting the PD-L1-lnc level, while half of beads were resuspended in RIPA lysis buffer for detecting the c-Myc protein level.

### RNA immunoprecipitation (RIP)

A549 cells transfected with PD-L1-lnc overexpressed vector or control vector were lysed with lysis buffer (20 mM Tris-HCl, 150 mM NaCl, 0.5% Nonidet P-40, 2 mM EDTA, 0.5 mM DTT, 1 mM NaF, 1 mM PMSF, 1% Protease Inhibitor Cocktail from Roche and 1000 U/ml Recombinant RNase Inhibitor from Takara, pH 7.4) for 30 min on ice. After cleared by centrifugation (13,000) for 20 min at 4 °C, the lysates were subjected to immunoprecipitation with anti-c-Myc antibody or IgG followed by protein A/G-Agarose beads. After the elution, the proteins were isolated by RIPA lysis buffer for western blot assays and the RNA were isolated with TRIzol™ Reagent for detecting PD-L1-lnc level.

### In vitro mRNA stability assay

To monitor the effect of PD-L1-lnc on the stability of PD-L1 mRNA, A549 cells transfected with control vector and PD-L1-lnc vector were treated by 5 μg/ml actinomycin D to block the transcription of new PD-L1 mRNA, and then assessed PD-L1 mRNA level by qRT-PCR at 0, 3, 6, 9, 12, or 24 h post-treatment.

### Statistical analysis

All experiments were performed in triplicate or as indicated in the experiments. Data were presented as the mean ± SD. When only two groups were compared, Student's *t* test was used. Comparisons involving multiple dependent measures were Tukey-Kramer corrected. The reported *P* value was 2-sided. The differences with *$P < 0.05$ were considered significantly different.

## Supplementary Information

---

**Additional file 1: Table S1.** Clinic characteristics of lung adenocarcinoma patients. **Table S2.** List of primers for qRT-PCR. **Table S3.** List of target sequences of various siRNAs.

**Additional file 2: Figure S1.** The PD-L1 protein level in human lung cancer tissue. **Figure S2.** The Sanger sequencing result of the PD-L1 mRNA (878 bp band) and PD-L1 lncRNA (705 bp band) obtained by RT-PCR. **Figure S3.** The Sanger sequencing result of the PD-L1 mRNA obtained by qRT-PCR with specific primer for amplification of PD-L1 mRNA (157 bp band) and PD-L1 lncRNA (106 bp band) obtained by RT-PCR. **Figure S4.** Expression of PD-L1-lnc in various cancers. **Figure S5.** PD-L1-lnc is a long non-coding RNA fragment. **Figure S6.** Induction of PD-L1 by IFNγ in lung adenocarcinoma cells. **Figure S7.** PD-L1-lnc has no effect on PD-L1 mRNA and protein expression. **Figure S8.** Expression level of genes within the c-Myc signaling downstream in lung cancer xenograft. **Figure S9.** PD-L1-lnc binds to c-Myc. **Figure S10.** Graphical representation images of the binding interface of the docking models of PD-L1-lnc with c-Myc by NPDock. **Figure S11.** The expression of PD-L1 lncRNA and mRNA in A549 cells transfected with the mutant vector and wild type vector. **Figure S12.** Cell fraction assay. **Figure S13.** Binding of

---

PD-L1-lnc to c-Myc enhances the association of c-Myc with MAX and promotes the entry of c-Myc into nucleus. **Figure S14.** The influence of c-Myc siRNA on gene expression in lung cancer A549 and PC9 cells. **Figure S15.** The effect of DAZAP1 on the generation of PD-L1-lnc.

**Additional file 3.** Source Data for blots. Uncropped Western blots and Agarose gels containing the entire ladder.

**Additional file 4.** Review history.

### Acknowledgements
The authors thank Professor Jill Leslie Littrell (Georgia State University, Atlanta, GA) for critical reading and constructive discussion of the manuscript.

### Review history
The review history is available as Additional file 4.

### Peer review information

### Authors' contributions
KZ, HL, and TW designed the research; SQ, ZJ, GL, BY, TW, WR, JX, TF, XS, and RY performed research and analyzed data; JW, YY, GX, XY, and TW collected the patients' samples; KZ and HL wrote the paper. All authors read and approved the final manuscript.

### Funding
This work was supported by grants from the Ministry of Science and Technology of China (2018YFA0507100), National Nature Science Foundation of China (31801088, 31670917, 31770981), Natural Science Foundation of Jiangsu Province (BK20170076) and Fundamental Research Funds for the Central Universities (020814380095, 020814380082).

### Availability of data and materials
The data of RNA sequencing analysis in this study can be viewed in GEO SRA database (PRJNA684685) [31]. The RNA-seq data and information of patients used in this study were obtained from TCGA database (https://www.cancer.gov/tcga.) [24].

## Declarations

### Ethics approval and consent to participate
This study was approved by the Ethics Committee of the Nanjing Drum Tower Hospital affiliated to Nanjing University School of Medicine, on the use of human samples for experimental studies. Written informed consent was obtained from each individual before enrolment. The Animal experiments were approved by the Ethics Committee of the Nanjing University School of Life Science. All the experiments were performed in accordance with the approved guidelines and complied with the Declaration of Helsinki.

### Competing interests
None declared.

### Author details
[1]State Key Laboratory of Pharmaceutical Biotechnology, School of Life Science, Nanjing University, Nanjing, China. [2]School of Life Science and Technology, China Pharmaceutical University, Nanjing, China. [3] Department of Thoracic Surgery, Nanjing Drum Tower Hospital, Medical School, Nanjing University    Nanjing   China   . [4]Department of Emergency Medicine, Nanjing Drum Tower Hospital, Medical School, Nanjing University, Nanjing, China. [5]Department of Medical Genetics, Nanjing Medical University, Nanjing, China. [6]Department of Pathology, Nanjing Drum Tower Hospital, Medical School, Nanjing University, Nanjing, China. [7]Department of General Surgery, Nanjing Drum Tower Hospital, Medical School, Nanjing University, Nanjing, China. [8]Department of Gastroenterology, Nanjing Drum Tower Hospital, Medical School, Nanjing University, Nanjing, China. [9]Department of Respiratory Medicine, Nanjing Drum Tower Hospital, Medical School, Nanjing University, Nanjing, China.

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

## 

