## [**Additional file 4.** Review history. · Genome Biology]

Review History

First round of review

Reviewer 1

Are you able to assess all statistics in the manuscript, including the appropriateness of statistical tests used? There are no statistics in the manuscript.

Comments to author:

Qu and colleagues present an impressive body of work to investigate the role of a possible noncoding transcript isoform of the PD-L1 locus. A range of functional and molecular biological evidence is presented to argue for PD-L1-lnc role in tumorigenesis, via its interaction with Myc. Surprisingly PD-L1-lnc does not seem to affect PD-L1 mRNA levels or protein.

Despite its strong points, I remain unsure about the significance of this work for a number of reasons:

- 1) There remains a distinct possibility that PD-L1-lnc encodes a peptide. This casts uncertainly on the conclusions of the entire paper, until resolved.
- 2) The paper initially raises interest in the possibility that a lncRNA could predict or be used to manipulate immunotherapy sensitivity. However this is not the case. PD-L1-lnc seems to be "just another cancer promoting lncRNA" (of which there are now hundreds known) - because there is no connection to PD-L1 protein expression or immunotherapy response. Therefore the impact of the work is rather limited.
- 3) The uncertainty of copy number of the transcript, and the odd choice to overexpress rather than knock down the transcript, also undermines confidence in the functional conclusions. Specific comments: Expression: The authors should comprehensively evaluate the FPKM values (RNA-seq) and copy number of PD-L1-lnc in the main cell models used here. It is curious that enough RNA is present for native RNA-antisense purification, and yet the authors choose to perform overexpression instead of knockdown to look for changing genes. Figure S4b- please give exact P value

Fig S5 - checking coding status. This experiment does not seem to make sense. The authors should use more rational approach to search for potential open reading frames (including using PhyloCSF track in UCSC genome browser, and ribosome profiling data) to evaluate the presence of ORFs. Its not sufficient to simply fuse the entire lncRNA to GFP. Given that the PD-L1-lnc overlaps so much of PD-L1 mRNA, the authors should produce a clear figure showing this alignment and where the ORF from mRNA overlaps the lncRNA.

Coding status: When I test NR_052005.2 in the NCBI ORF finder software, it returns a 178 aa peptide prediction, below. Please comment. The authors might clone this sequence into an overexpression vector and perform Western for the predicted peptide.

>|cl|ORF1

MRIFAVFIFMTYWHLNNAFTVTVPKDLYVVEYGSNMTIECKFPVEKQLDL

AALIVYWEMEDKNIIQFVHGEEEDLKVQHSSYRQRARLLKDQLSLGNAALQ

ITDVKLQDAGVYRCMISYGGADYKRITVKVNAPYNKINQRILVVDPVTSE

HELTCQAEGYPKAEVIWTSSDHQVLSGD

Figure 6A: Its curious why overexpression instead of knockdown was used to assess function. KD would be preferable, revealing genuine effects and not artefacts of overexpression. The authors should estimate the number of copies of PD-L1-lnc in the overexpression and natural situations.

Reviewer 2

Are you able to assess all statistics in the manuscript, including the appropriateness of statistical tests used? Yes, and I have assessed the statistics in my report.

Comments to author:

Authors describe how a non protein coding transcript, PD-L1-lnc, arises from of the alternative splicing of PD-L1 mRNA in lung adenocarcinoma tumor samples. This transcript is expressed at similar levels in PD-L1 protein negative and positive tumor samples and promotes lung cancer cell proliferation in vitro and in vivo. The authors then demonstrate that PD-L1-lnc does not exert its function through the protection of PD-L1 mRNA from nonsense-mediated decay but conveys its pro-tumoral properties through its interaction with c-Myc. Authors provide data suggesting that PD-L1-lnc facilitates helix-loop-helix Zip motif formation in c-Myc and demonstrate increased heterodimer formation with protein chaperone Max when PD-L1-lnc is overexpressed in A549 cells. Finally authors demonstrate that c-Myc entry into the nucleus of lung adenocarcinoma cell model A549 is increased when PD-L1-lnc is overexpressed and that its pro-tumoral properties are c-Myc dependent.

This work offers a convincing data and sheds light on a new c-Myc regulatory role for lncRNAs in lung cancer. I would recommend the article be accepted for publication given that authors address the following points.

Major Points

1° The observations made in regards to the PD-L1-lnc and c-Myc axis would gain if authors reproduced their findings in other PD-L1 protein negative cell models.

2° Authors show increased c-Myc immunoprecipitation when PD-L1-lnc is overexpressed and increased nuclear distribution of c-Myc. Given that depletion or overexpression of PD-L1-lnc down- or upregulates c-Myc target genes it would be of interest to show by ChIP-qPCR enriched or decreased c-Myc presence at deregulated loci. Another solution would be to perform qPCR for c-Myc gene panel (showed in fig.6C) in PD-L1-lnc overexpressing cells with c-Myc siRNA.

3° Figure 6C, statistical data should be shown for c-Myc panel of genes. Given the small fold-change range indicated it would be necessary to provide significance for displayed genes.

4° Figure 7 F-G, authors show with IF the distribution of PD-L1 protein in A549 cells in control vector or PD-L1-lnc overexpressing vector. It appears that the overexpression of PD-L1-lnc leads to a striking increase of cytoplasmic c-Myc towards the nucleus. These results are further supported by western blot analysis (Figure 7E). It is unclear what method/software was used to quantify relative protein levels and if bands are normalized to loading control.

Furthermore, these findings would gain in strength if authors showed the effect of overexpressing c-Myc binding domain mutant PD-L1-lnc and the effect of PD-L1-lnc depletion, as a considerable amount of c-Myc is observed in control vector condition. Value would also be added to these results if Max protein distribution were also show by IF as one wonders where the ectopically expressed PD-L1-lnc meets c-Myc and Max ? Authors should therefore provide PD-L1-lnc fractionation data when overexpressing PD-L1-lnc.

Minor Points

1° Page 3, line 4 "sophistic" should be "sophisticated"

2° Page 13, line 47 "no correction" should be "no correlation"

3° Figure S7 B, legends states it is RNA expression are they also normalized by GAPDH ?

4° Figure S7 C and D, what the composition of these cohorts only PD-L1 protein negative or do they contain both PD-L1 protein positive and negative tumor samples ? What is the cut-off used to categorize samples within high or low expression groups ? How many tumors fall within each group ? Authors should provide statistical data and explain If plotted data is relapse-free or overall survival data.

Authors Response

Point-by-point responses to the reviewers' comments:

Reviewer 1:

1) There remains a distinct possibility that PD-L1-lnc encodes a peptide. This casts uncertainly on the conclusions of the entire paper, until resolved.

Response: We agree with reviewer that whether PD-L1-lnc encodes a peptide is a critical issue. In fact, we test this through various experiments. NCBI ORF finder software predicts that PD-L1-lnc may encode a 178aa peptide which shows a significant similarity to the PD-L1 protein. According to the reviewer's suggestion, we cloned PD-L1-lnc sequence, as well as PD-L1 mRNA into the overexpression vector (pcDNA3.1) with a His-tag (new Supplemental Fig. S5C). As shown in new Supplemental Fig. S5D, no peptide encoded by PD-L1-lnc sequence was detected by anti-His antibody, whereas a peptide encoded by PD-L1 mRNA was detected. As this predicted peptide is almost identical to PD-L1 protein, it should be detected by PD-L1 antibody against the same protein region. However, when we transfected cells with PD-L1-lnc, we did not detect such peptide using several anti-PD-L1 antibody (new Supplemental Fig. S5D, upper). In our original submission, we also linked PD-L1-lnc to GFP mRNA in an expressing vector (new Supplemental Fig. S5A). RT-PCR assay and fluorescence microscopy showed a high level of GFP mRNA (new Supplemental Fig. S5A, lower) but no GFP protein in the PD-L1-lnc-expressing system (new Supplemental Fig. S5B). These results confirm that PD-L1-lnc does not encode a protein but functions as a lncRNA. Supporting this conclusion, analysis of ribosome profiling data of A549 cells collected in NCBI GEO database (GSE129654) also failed to identify PD-L1-lnc sequence. We have also revised the manuscript accordingly (Page 13-14, the changed parts are marked in red).

2) The paper initially raises interest in the possibility that a lncRNA could predict or be used to manipulate immunotherapy sensitivity. However this is not the case. PD-L1-lnc seems to be "just another cancer promoting lncRNA" (of which there are now hundreds known) -

because there is no connection to PD-L1 protein expression or immunotherapy response. Therefore the impact of the work is rather limited.

Response: We understand reviewer's concern on this issue. In fact, as reviewer mentioned, we started our study aiming to illustrate the protection of PD-L1-lnc on PD-L1 protein through the mechanism of nonsense-mediated mRNA decay (NMD). However, extensive studies in this manuscript show that PD-L1-lnc does not regulate PD-L1 protein level. Although there is no direct connection to PD-L1 protein expression or immunotherapy response, our results show that PD-L1-lnc has a significant protumorigenic function independent of PD-L1 protein. Both in vitro and in vivo data demonstrate that PD-L1-lnc executes its cancer-promoting role through binding to cMyc and increasing Myc-Max signal axis. Given the importance of PD-L1 as an immune checkpoint and the limitation of PD-L1 blockade in solid tumor immunotherapy, we believe that identification of an independent cancer-promoting role of PD-L1-lnc has a significance in anti-cancer treatment. As production of PD-L1-lnc is not dependent upon PD-L1 mRNA/protein but also upregulated by IFN γ , PD-L1-lnc is not just another cancer-promoting lncRNA, but an important molecule underlying the limitation of PD-L1 blockade in tumor immunotherapy. Our studies strongly argue that blocking PD-L1 to enhance T cell immunity may be not enough to suppress tumor progression particularly the metastasis, and in addition to PD-L1 blockade, depleting PD-L1-lnc in tumor cells should be considered in order to prevent tumor progression.

3) The uncertainty of copy number of the transcript, and the odd choice to overexpress rather than knock down the transcript, also undermines confidence in the functional conclusions.

Response: We appreciate reviewer's comment on this. Accordingly, we performed RNA sequencing in A549 cells and found that the copy number of PD-L1-lnc was about 10 FPKM (GEO SRA database: PRJNA684685). We also depleted the PD-L1-lnc using PD-L1-lnc-specific shRNA and investigated the function of PD-L1-lnc. For example, we performed EdU staining, Transwell assay and Annexin V labeling in A549 and PC9 lung cancer cells following PD-L1-lnc knockdown. The results showed that the proliferation (new Fig. 4A) and invasion (new Fig. 4B) of cancer cells were significantly reduced, whereas the apoptosis of cancer cells (new Fig. 4C) was increased when PDL1-lnc was depleted via specific PD-L1-lnc shRNA. In new Figure 5, we also evaluated the influence of PD-L1-lnc on lung cancer progression using mouse models. The results showed that A549 cells with PD-L1-lnc depletion had a growth delay compared to control A549 cells. In new Figure 6, we also investigated the gene expression following either overexpression or knockdown of PD-L1-lnc. We have also revised the manuscript accordingly (Page 16-17, the changed parts are marked in red).

Specific comments:

Expression: The authors should comprehensively evaluate the FPKM values (RNA-seq) and copy number of PD-L1-lnc in the main cell models used here. It is curious that enough RNA is present for native RNA-antisense purification, and yet the authors choose to perform overexpression instead of knockdown to look for changing genes.

Response: We appreciate reviewer's constructive comments, and we have addressed these issues in our response to question 3.

Figure S4b- please give exact P value.

Response: We added the exact p value in the Figure S4B, in which the p value was 0.0083.

Fig S5 - checking coding status. This experiment does not seem to make sense. The authors should use more rational approach to search for potential open reading frames (including using PhyloCSF track in UCSC genome browser, and ribosome profiling data) to evaluate the presence of ORFs. Its not sufficient to simply fuse the entire lncRNA to GFP. Given that the PD-L1-lnc overlaps so much of PD-L1 mRNA, the authors should produce a clear figure showing this alignment and where the ORF from mRNA overlaps the lncRNA.

Coding status: When I test NR_052005.2 in the NCBI ORF finder software, it returns a 178 aa peptide prediction, below. Please comment. The authors might clone this sequence into an overexpression vector and perform Western for the predicted peptide.

>|c|ORF1

MRIFAVFIFMTYWHLNAFTVTVPKDLYVVEYGSNMTIECKFPVEKQLDL
AALIVYWEMEDKNIIQFVHGEEEDLKVQHSSYRQRARLLKDQLSLGNAALQ
ITDVKLQDAGVYRCMISYGGADYKRITVKVNAPYNKINQRILVVDPTSE
HELTCQAEGYPKAEVIWTSSDHQVLSGD

Response: We thank reviewer for pointing out this critical issue, and we have addressed this in our response to question 1.

Figure 6A: Its curious why overexpression instead of knockdown was used to assess function. KD would be preferable, revealing genuine effects and not artefacts of overexpression. The authors should estimate the number of copies of PD-L1-lnc in the overexpression and natural situations.

Response: Indeed, we agree with reviewer that knockdown is a better way (compared to overexpression) to reveal genuine effects of PD-L1-lnc since tumor cells already have a considerable PD-L1-lnc expression. Accordingly, we performed RNA sequencing for A549 cells and found that PD-L1-lnc copy number was about 10.88FPKM, 102.55 FPKM or 1.45 FPKM in A549 cells transfected with control vector, PD-L1-lnc-expressing vector (PD-L1-lnc) or PD-L1-lnc shRNA, respectively (GEO SRA database: PRJNA684685). We also investigated the gene expression regulated by PDL1-lnc in tumor cells following either overexpression or knockdown of PD-L1-lnc. As shown in new Figure 6A, the genes that were upregulated by PD-L1-lnc overexpression were found to be downregulated when PD-L1-lnc was knocked down by PD-L1-lnc shRNA. We have also revised the manuscript accordingly (Page 17, the changed parts are marked in red).

Reviewer 2:

1. The observations made in regards to the PD-L1-lnc and c-Myc axis would gain if authors reproduced their findings in other PD-L1 protein negative cell models.

Response: We greatly appreciate reviewer's constructive suggestion. Accordingly, we selected MCF7 (breast cancer cell line) and Huh7 cells (liver cancer cell line), two cancer

cell lines expressed little or no PD-L1 protein. We performed coimmunoprecipitation in MCF7 and Huh7 cells, and as shown in new Supplemental Fig. S8, we obtained the same results as in the A549 cells. We have also revised the manuscript accordingly (Page 17, the changed parts are marked in red).

2. Authors show increased c-Myc immunoprecipitation when PD-L1-lnc is overexpressed and increased nuclear distribution of c-Myc. Given that depletion or overexpression of PD-L1-lnc down- or upregulates c-Myc target genes it would be of interest to show by ChIP-qPCR enriched or decreased c-Myc presence at deregulated loci. Another solution would be to perform qPCR for c-Myc gene panel (showed in fig.6C) in PD-L1-lnc overexpressing cells with c-Myc siRNA.

Response: We greatly appreciate the reviewer's constructive suggestion. Accordingly, we performed qPCR for c-Myc gene panel (new Fig. 6C) in PD-L1-lnc overexpressing cells with c-Myc siRNA. As shown in new Fig. 6C, the c-Myc siRNA markedly attenuated the promotion of these genes by PD-L1-lnc overexpression (Page 17, the changed parts are marked in red).

3. Figure 6C, statistical data should be shown for c-Myc panel of genes. Given the small fold-change range indicated it would be necessary to provide significance for displayed genes.

Response: According to reviewer's comment, we provided the significance for displayed genes in new Fig. 6C.

4. Figure 7 F-G, authors show with IF the distribution of PD-L1 protein in A549 cells in control vector or PD-L1-lnc overexpressing vector. It appears that the overexpression of PD-L1-lnc leads to a striking increase of cytoplasmic c-Myc towards the nucleus. These results are further supported by western blot analysis (Figure 7E). It is unclear what method/software was used to quantify relative protein levels and if bands are normalized to loading control. Furthermore, these findings would gain in strength if authors showed the effect of overexpressing c-Myc binding domain mutant PD-L1-lnc and the effect of PD-L1-lnc depletion, as a considerable amount of c-Myc is observed in control vector condition. Value would also be added to these results if Max protein distribution were also show by IF as one wonders where the ectopically expressed PD-L1-lnc meets c-Myc and Max ? Authors should therefore provide PD-L1-lnc fractionation data when overexpressing PD-L1-lnc.

Response: We employed Image J to quantify relative protein levels and the bands were normalized to loading control (input). According to the reviewer's suggestion, we also showed the effect of overexpressing c-Myc binding domain mutant PD-L1-lnc (PDL1-lnc mut) and the effect of PD-L1-lnc depletion (PD-L1-lnc del) in new Supplemental Figure S11. According the RNA sequencing for A549 cells, we found that PD-L1-lnc copy number was about 10.88 FPKM in A549 cells, while it increased to 102.55 FPKM in A549 cells after transfection with PD-L1-lnc-expressing vector (PD-L1-lnc) (GEO SRA database: PRJNA684685). Moreover, we also investigated the expression level of PD-L1-lnc and Max protein distribution in A549 cells transfected with PD-L1-lnc-expressing vector (PD-L1-lnc). As the results shown in new Supplemental Fig. S10A, the PD-L1-lnc significantly increased in A549 cells transfected with PD-L1-lnc-expressing vector (PD-L1-lnc). Moreover, Immunoprecipitation and Western blot analysis further indicated that overexpression of PD-L1-lnc in A549 cells mainly enhanced the nuclear distribution of Max which were associated with c-Myc, while knockdown of PD-L1-lnc in A549 cells suppressed the nuclear distribution of Max (new Fig. 7E-G). However, neither overexpression nor knockdown of PD-L1-lnc in

A549 cells has effect on the total cellular level of c-Myc-associated Max (new Fig. 7E-G). We have revised the manuscript accordingly (Page 15, the changed parts are marked in red).

Minor Points

1. Page 3, line 4 "sophistic" should be "sophisticated".

Response: We have corrected the "sophistic" to "sophisticated".

2. Page 13, line 47 "no correction" should be "no correlation".

Response: We have corrected the "no correction" to "no correlation".

3. Figure S7 B, legends states it is RNA expression are they also normalized by GAPDH?

Response: The RNA expression in Figure S7B is normalized by Transcripts Per Kilobase of exon model per Million mapped reads (TPM).

4. Figure S7 C and D, what the composition of these cohorts only PD-L1 protein negative or do they contain both PD-L1 protein positive and negative tumor samples ? What is the cut-off used to categorize samples within high or low expression groups ? How many tumors fall within each group ? Authors should provide statistical data and explain If plotted data is relapse-free or overall survival data.

Response: The data in the Figure S7, C-D were obtained from TCGA database, and indeed, these samples contained both PD-L1 protein positive and negative tumor samples. The cut-off used to categorize samples within high or low expression groups was 3.90TPM. For the high expression groups, we obtained 246 samples. For the low expression groups, we obtained 147 samples. The statistical data was added in the new Supplemental Figure S7, C and D. In statistical analysis, the plotted data is overall survival data. We have revised the manuscript accordingly (Page 15, the changed parts are marked in red).

Second round of review

Reviewer 1

My concerns have been allayed.

Reviewer 2

MAJOR POINTS

1. While authors have satisfactory addressed comments regarding overexpression in their new experiments, they do not integrate enough the striking different intensities in the effect of PDL1nc depletion in comparison to overexpression in their results and discussion. Depletion doesn't exactly lead to a mirrored effect of overexpression. Authors should make an effort to tone down their conclusions given the the milder effects of depletion. Authors should further discuss the biological relevance of lowly expressed lncRNA transcripts and provide evidence that such low abundant transcript have been described as functional in other contexts to support the relevance of PDL1nc. While lncRNA are usually less abundant than coding genes, 10 FPKM is rather of low side of less abundant molecules.

2. When describing the RNA-seq data for shRNA or overexpressing PDL1inc, in the "PD-L1- Inc executes its oncogenic function through activating c-Myc signaling" section, it is unclear what the cut-off for defining down or upregulated genes is nor how many genes were indeed deregulated in total. Moreover the material and method section fails to provide analysis parameters ? What did authors use to map reads ? To perform differential gene expression analysis ?

3. In their answer to my point regarding c-MYC and Max distribution, authors provide valuable data supporting their claims for c-Myc but do not provide fractionation data on PDL1inc, therefore I would advise authors to provide a simple qPCR of PDL1inc in from cell fractions or blot as in figure 3C but for the PDL1 transfected cells. If indeed c-Myc is affected by PDL1inc overexpression then its levels should accompany those of c-Myc.

4. The material and method section should provide a more detailed section on A549 models, pcDNA3 backbone is not the most appropriate for stable integration so if users did indeed use this backbone then procedure should be more detailed. Same for stable shRNA generation, were lentiviral particles generated in HEK293 ? How many cells were seeded, what was the MOI of infection ? how long were the cells selected. It is unclear if lipofectamine was used in this case but usually lentiviral particles are delivered with polybrene. The material and method should provide a section on how authors performed cellular fractionation.

5. Finally, authors should change their final comment " and provides targeting PD-L1- Inc-c-Myc axis as a novel therapeutic strategy to improve the efficacy of lung cancer therapy." to something more suggestive such as " results argue in favor of investigating PDL1inc depletion in combination with PDL1 therapy" OR provide data showing that indeed depleting PDL1inc enhances PD1/PD-L1 therapy.

MINOR POINTS

Figure 6: A/ what does this heatmap show ? What is the scale on the left, fold-change or log FC ? The small range (from -1 to 1) doesn't agree with the qPCR shown in figure 6C where 8-fold differences are observed, perhaps this heatmap should only present the genes validated by qPCR, it would be clearer and more informative. B/ Authors should annotate what the percentages right of the bars reflect. C/ Impact of PDL1inc depletion and overexpression should be presented separately as the bigger impact of over-expression makes the effect of depletion unclear to assess.

Figure 7: Provide a more detailed legend and annotate better the figure. Authors should explain that in 7D, c-Myc doesn't bind GFPmRNA but the PDL1inc-GFP transcript and that the upper 106bp band is the endogenous transcript.

In their answer to my point regarding c-MYC and Max distribution, authors provide valuable data supporting their claims for c-Myc but do not provide fractionation data on PDL1inc, therefore I would advise authors to provide a simple qPCR of PDL1inc in from cell fractions or blot as in figure 3C but for the PDL1 transfected cells. If indeed c-Myc is affected by PDL1inc overexpression then its levels should accompany those of c-Myc.

Figure S8 : PD-L1inc bands for MCF7 and Huh cells are exactly the same the shape, I overlapped them and they exactly matched, although bands can appear similar, I've never

been able to exactly map bands, I would advise authors to verify if the same blot was not mistakenly used twice.

Authors Response

Point-by-point responses to the reviewers' comments:

Point-to-point response

Reviewer #2:

MAJOR POINTS

1. *While authors have satisfactorily addressed comments regarding overexpression in their new experiments, they do not integrate enough the striking different intensities in the effect of PDL1inc depletion in comparison to overexpression in their results and discussion. Depletion doesn't exactly lead to a mirrored effect of overexpression. Authors should make an effort to tone down their conclusions given the milder effects of depletion. Authors should further discuss the biological relevance of lowly expressed lncRNA transcripts and provide evidence that such low abundant transcripts have been described as functional in other contexts to support the relevance of PDL1inc. While lncRNAs are usually less abundant than coding genes, 10 FPKM is rather of low side of less abundant molecules.*

Response: We appreciate reviewer's comments. Firstly, we have toned down our conclusions according to reviewer's advice in the revised manuscript (Page 2 and 24, the changed parts are marked in red). Secondly, we also discussed the biological relevance of lowly expressed lncRNA transcripts and provide evidence that such low abundant lncRNA transcripts have been described as functional active in other previous reports to support the relevance of PD-L1-lnc in the discussion section (Page 22, the changed parts are marked in red).

2. *When describing the RNA-seq data for shRNA or overexpressing PDL1inc, in the "PD-L1-lnc executes its oncogenic function through activating c-Myc signaling" section, it is unclear what the cut-off for defining down or upregulated genes is nor how many genes were indeed deregulated in total. Moreover the material and method section fails to provide analysis parameters? What did authors use to map reads? To perform differential gene expression analysis?*

Response: We thank reviewer for pointing out these critical issues. Accordingly we have provided the analysis parameters in the material and method section in our revised manuscript (Page 10, the changed parts are marked in red). To calculate the expression of the transcripts, Cutadapt software (<https://cutadapt.readthedocs.io/en/stable/>, version: cutadapt-1.9) was used to remove the reads that contained adaptor contamination. After removal of the low quality and undetermined bases, HISAT2 software (<https://daehwankimlab.github.io/hisat2/>, version: hisat2-2.0.4) was used to map reads to the genome. The mapped reads of each sample were assembled using StringTie (<http://ccb.jhu.edu/software/stringtie/>, version: stringtie-1.3.4d.Linux_x86_64) with default parameters. Then, all transcriptomes from all samples were merged to reconstruct a comprehensive transcriptome using gffcompare software (<http://ccb.jhu.edu/software/stringtie/gffcompare.shtml>, version: gffcompare-0.9.8.Linux_x86_64). After the final transcriptome was generated, StringTie and ballgown (<http://www.bioconductor.org/packages/release/bioc/html/ballgown.html>) were used to estimate the expression levels of all transcripts and perform expression level for mRNAs by calculating FPKM ($FPKM = [\text{total exon fragments} / \text{mapped reads (millions)} \times \text{exon length(kB)}]$). The differentially expressed mRNAs were selected with fold change > 2 or fold change < 0.5 by R package edgeR (<https://bioconductor.org/packages/release/bioc/html/edgeR.html>). We found 699 dysregulated genes (350 genes fold change > 2 in A549 cells transfected with PD-L1-lnc-expressing vector (PD-L1-lnc) compared to A549 cells transfected with control vector (CTL vector), and fold change < 0.5 in A549 cells transfected with PD-L1-lnc shRNA vector (PD-L1-lnc shRNA) compared to A549 cells transfected with control vector (CTL vector); 349 genes fold change < 0.5 in A549 cells transfected with PD-L1-lnc-expressing vector (PD-L1-lnc) compared to A549 cells transfected with control vectors (CTL vector) and fold change > 2 in A549 cells transfected with PD-L1-lnc shRNA vector (PD-L1-lnc shRNA) compared to A549 cells transfected with control

vectors (CTL vector)).

3. In their answer to my point regarding *c-MYC* and Max distribution, authors provide valuable data supporting their claims for *c-Myc* but do not provide fractionation data on *PDL1*inc, therefore I would advise authors to provide a simple qPCR of *PDL1*inc in from cell fractions or blot as in figure 3C but for the *PDL1* transfected cells. If indeed *c-Myc* is affected by *PDL1*inc overexpression then its levels should accompany those of *c-Myc*.

Response: We have provided the blot of PD-L1-*inc* in cell fractions according to the reviewer's suggestion. As shown in new Supplemental Fig. S11, the PD-L1-*inc* was accompanied the distribution of *c-Myc* in the cell fraction.

4. The material and method section should provide a more detailed section on A549 models, pcDNA3 backbone is not the most appropriate for stable integration so if users did indeed used this backbone then procedure should more detailed. Same for stable shRNA generation, were lentiviral particles generated in HEK293? How many cells were seeded, what was the moiety of infection? how long were the cells selected. It is unclear if lipofectamine was used in this case but usually lentiviral particles are delivered with polybrene. The material and method should provide a section on how authors performed cellular fractionation.

Response: We appreciate reviewer's comment on this. Detailed information about cell seeding and infection was provided in the material and method section of our revised manuscript (Page 5-6 and 8, the changed parts are marked in red).

5. Finally, authors should change their final comment "and provides targeting PD-L1-*inc*-*c-Myc* axis as a novel therapeutic strategy to improve the efficacy of lung cancer therapy." to something more suggestive such as "results argue in favor of investigating *PDL1*inc depletion in combination with *PDL1* therapy" OR provide data showing that indeed depleting *PDL1*inc enhances *PD1/PD-L1* therapy.

Response: We have revised our conclusion in the revised manuscript according to reviewer's suggestion (Page 2 and 24, the changed parts are marked in red).

MINOR POINTS

Figure 6: A/ what does this heatmap show? What is the scale on the left, fold-change or log FC? The small range (from -1 to 1) doesn't agree with the qPCR shown in figure 6C where 8-fold difference are observed, perhaps this heatmap should only present the genes validated by qPCR, it would be clearer and more informative. B/ Authors should annotate what the percentages right of the bars reflect. C/ Impact of *PDL1*inc depletion and overexpression should be presented separately as the bigger impact of over-expression makes the effect of depletion unclear to assess.

Response: We apologize for wrong scale bar label in the Figure 6A and we have changed it accordingly. The Figure 6A showed the heatmap of dysregulated genes (350 genes fold change > 2 in A549 cells transfected with PD-L1-*inc* overexpression vector (PD-L1-*inc*) compared to A549 cells transfected with control vectors (CTL shRNA) and fold change < 0.5 in A549 cells transfected with PD-L1-*inc* shRNA vector (PD-L1-*inc* shRNA) compared to A549 cells transfected with control vectors (CTL vector); 349 genes fold change <0.5 in A549 cells transfected with PD-L1-*inc* overexpression vector (PD-L1-*inc*) compared to A549 cells transfected with control vectors (CTL vector) and fold change >2 in A549 cells transfected with PD-L1-*inc* shRNA vector (PD-L1-*inc* shRNA) compared to A549 cells transfected with control vectors (CTL vector)). For Figure 6B, the bars indicate

the percent of dysregulated genes regulated by the transcriptional factor. For Figure 6C, PD-L1-lnc depletion and overexpression in A549 cells have been presented separately, and the results derived from mice have been moved to new Supplemental Fig. S8.

Figure 7: Provide a more detailed legend and annotate better the figure. Authors should explain that in 7D, c-Myc doesn't bind GFPmRNA but the PDL1lnc-GFP transcript and that the upper 106bp band is the endogenous transcript.

Response: We appreciate reviewer's comment on this. As shown in the figure below, the A549 cells were transfected with PD-L1-lnc Vector (which expresses GFP/PD-L1-lnc), PD-L1-lnc Mut Vector (which expresses GFP/PD-L1-lnc with mutated c-Myc binding site), PD-L1-lnc Del Vector (which expresses GFP/PD-L1-lnc with deleted c-Myc binding site) or CTL Vector. A549 cells transfected with CTL Vector only produced endogenous PD-L1-lnc in the cells, while A549 cells transfected with PD-L1-lnc Vector generated both endogenous PD-L1-lnc and GFP/PD-L1-lnc. For A549 cells transfected with PD-L1-lnc Mut Vector or PD-L1-lnc Del Vector, cells produced not only PD-L1-lnc, but also GFP/PD-L1-lnc with mutated c-Myc binding site or deleted c-Myc binding site. When the c-Myc protein was pulled down with anti-c-Myc antibody, there was only endogenous PD-L1-lnc in the immunoprecipitation complex from A549 cells transfected with CTL Vector. However, for A549 cells transfected with PD-L1-lnc Vector, both endogenous PD-L1-lnc and GFP/PD-L1-lnc were immunoprecipitated. For A549 cells transfected with PD-L1-lnc Mut Vector or PD-L1-lnc Del Vector, there was only endogenous PD-L1-lnc but no GFP/PD-L1-lnc or GFP/PD-L1-lnc Del transcripts in the immunoprecipitated complex since the c-Myc binding site on PD-L1-lnc was mutated or deleted, respectively. Employing PD-L1-lnc primer, we are able to detect the endogenous PD-L1-lnc and the exogenous PD-L1-lnc expressed by the vectors in the immunoprecipitation complex from A549 cells transfected with PD-L1-lnc Vector. In contrast, employing GFP primer, we only detected the GFP/PD-L1-lnc in the immunoprecipitation complex from A549 cells transfected with PD-L1-lnc Vector.

Figure S8 : PD-L1 Inc bands for MCF7 and Huh cells are exactly the same the shape, I overlapped them and they exactly matched, although bands can appears similar, I've never been able to exactly map bands, I would advise authors to verify if the same blot was not mistakenly used twice.

Response: We appreciate reviewer's comments/suggestions. We have carefully verified the results and confirmed we didn't mistakenly used twice. We provided the raw gel below. In fact, all uncropped blots containing the entire ladder were submitted as supplementary figures.

Third round of review

Reviewer 2

Satisfactory answer to all comments.

Minor points:

Line 269: “However, despite at low expression level....” Should be “despite their low expression level

Line 580: “ However, despite at the relative low level of expression, ...” the “at” should be removed.